# Histone H3.3 lysine 9 and 27 control repressive chromatin at cryptic enhancers and bivalent promoters

Matteo Trovato [1,2], Daria Bunina[1,3], Umut Yildiz [1,2], Nadine Fernandez-Novel Marx[1], Michael Uckelmann[4], Vita Levina[4], Yekaterina Perez [5], Ana Janeva[1], Benjamin A. Garcia [5], Chen Davidovich [4], Judith B. Zaugg [3] & Kyung-Min Noh [1] ✉

Histone modifications are associated with distinct transcriptional states, but it is unclear whether they instruct gene expression. To investigate this, we mutate histone H3.3 K9 and K27 residues in mouse embryonic stem cells (mESCs). Here, we find that H3.3K9 is essential for controlling specific distal intergenic regions and for proper H3K27me3 deposition at promoters. The H3.3K9A mutation resulted in decreased H3K9me3 at regions encompassing endogenous retroviruses and induced a gain of H3K27ac and nascent transcription. These changes in the chromatin environment unleash cryptic enhancers, resulting in the activation of distinctive transcriptional programs and culminating in protein expression normally restricted to specialized immune cell types. The H3.3K27A mutant disrupts the deposition and spreading of the repressive H3K27me3 mark, particularly impacting bivalent genes with higher basal levels of H3.3 at promoters. Therefore, H3.3K9 and K27 crucially orchestrate repressive chromatin states at *cis*-regulatory elements and bivalent promoters, respectively, and instruct proper transcription in mESCs.

Specific histone PTMs are associated with gene expression through active or repressive chromatin states. For instance, histone H3 lysine 9 and lysine 27 acetylation (H3K9ac and H3K27ac) mark active *cis*-regulatory elements (CREs), whereas tri-methylation of these residues is a repressive mark, found at silenced repetitive elements (H3K9me3) and gene promoters (H3K27me3)[1–3]. Although a strong association between these PTMs and gene expression has been established[4,5], a systemic study to investigate the precise contribution of specific histone residues and associated PTMs to the regulation of chromatin environments and gene expression is still lacking in mammals.

Histone H3 is present as three variants, H3.1-3. Canonical histone H3 (H3.1/H3.2) is produced from multiple gene copies and incorporated into nucleosomes upon DNA synthesis, thus being evenly distributed across the genome. In contrast, the incorporation of histone H3.3 is replication-independent. H3.3 differs from H3.1 and H3.2 by 5 and 4 amino acids, respectively[6]. The H3.3-specific residues are recognized by H3.3-specific chaperones, which deposit H3.3 at specific genomic regions, including actively transcribed genes and *cis*-regulatory elements, during interphase[7–11]. Histone H3.3 represents only ~20% of the total H3 pool in proliferating cells[12]; however, given that H3.3 is enriched at promoters, enhancers, repeat elements, and active

[1]European Molecular Biology Laboratory (EMBL), Genome Biology Unit, Heidelberg, Germany. [2]Collaboration for joint PhD degree between EMBL and Heidelberg University, Faculty of Biosciences, Heidelberg, Germany. [3]European Molecular Biology Laboratory (EMBL), Structural and Computational Biology Unit, Heidelberg, Germany. [4]Department of Biochemistry and Molecular Biology, Biomedicine Discovery Institute, Faculty of Medicine, Nursing and Health Sciences, Monash University, and EMBL-Australia, Clayton, VIC, Australia. [5]Department of Biochemistry and Molecular Biophysics, Washington University School of Medicine, St. Louis, MO, USA. ✉e-mail: kyung.min.noh@embl.de

genes, mutations of H3.3 result in substantial epigenome changes in mammalian cells and contribute to the pathogenesis of cancers[13–19]. Histone H3.3 is encoded by two genes (*H3f3a* and *H3f3b*), not located within the main histone gene clusters. The *H3f3a* and *H3f3b* double-knockout (KO) is embryonic lethal in mice, while single KOs are viable[20–22].

In this study, we investigate the role of H3.3K9 and H3.3K27 in transcriptional regulation and chromatin response by conducting histone H3.3 gene mutagenesis in mESCs[18,19]. Loss of function lysine-to-alanine mutations in either the K9 or K27 residue results in significant transcriptional perturbation in mutant mESCs, accompanied by proliferation and/or differentiation defects. We find that H3.3K9 plays a crucial role in preserving specific heterochromatic regions and regulating cryptic *cis*-regulatory elements derived from endogenous retroviruses (ERVs). An mESC-specific gene regulatory network reveals a connection between activated ERV-derived CREs and immune genes, as well as a subset of bivalent genes, that are selectively upregulated in H3.3K9A mutant cells. In contrast, H3.3K27 is directly required for transcriptional repression of a distinct subset of bivalent genes as their promoters are enriched with H3.3 and H3K27me3. These results give evidence for an instructive role of H3.3K9 and K27 residues in the regulation of gene expression. Our study also provides mechanistic insights into how H3.3K9 contributes to maintaining defined heterochromatic regions and controlling the expression of associated genes.

## Results

### H3.3K9A and K27A perturb transcription in mESCs

To elucidate the function of H3.3K9 and H3.3K27 and their modifications, we substituted K9 or K27 for alanine (H3f3a$^{-/-}$; H3f3b$^{K9A/K9A}$ or H3f3a$^{-/-}$; H3f3b$^{K27A/K27A}$) in mESCs and compared them to control mESCs (H3f3a$^{-/-}$; H3f3b$^{WT/WT}$) (Fig. 1a and Supplementary Fig. 1). Hereafter, we refer to these lines as K9A, K27A, and control mESCs. For each mutant, we generated three independent biological replicates, and after validating the chromosomal integrity of the edited lines (Supplementary Fig. 1f), we assessed changes in the global levels of histone modifications and gene expression.

Histone mass-spectrometry (MS) of the control and mutant lines confirmed the K9A and K27A substitutions in H3.3 proteins (Supplementary Fig. 2a). MS also revealed the effects of the H3.3 mutations on global histone modification levels in H3.3 and H3.1/H3.2. For instance, we observed increased H3.1/H3.2K27me1 levels but unaltered H3.1/H3.2K27me2/3 levels in K27A mESCs compared to the control, indicating no compensatory effect on H3.1/H3.2K27me3 upon H3.3K27A substitution (Supplementary Fig. 2b). In contrast, K9A mESCs showed decreased H3.3K27me1/2/3 levels and a reduction in H3.1/H3.2K27me1/2/3 levels as well as increases in histone H3.1/H3.2/H3.3 (H3K9ac, H3K14ac, H3K18ac, and H3K23ac) and H4 acetylation (Supplementary Fig. 2a–c), indicating broad changes in chromatin modifications upon H3.3K9 substitution.

To assess changes in gene expression, we performed mRNA-seq. Hierarchical clustering of the normalized read counts showed distinct transcriptional profiles for K9A and K27A cells, compared to controls (Fig. 1b). Analysis of differentially expressed genes (DEGs) revealed substantial changes for K9A (2585 DEGs, *p*adj <0.05 & FC cut-off = 2) and K27A mESCs (1463 DEGs, *p*adj <0.05 & FC cut-off = 2), with the number of DEGs and the magnitude of gene expression changes being greater in K9A than K27A (Fig. 1c, d).

To further characterize the gene expression changes in each mutant, we performed gene ontology (GO) enrichment analysis on the DEGs (Fig. 1e). Upregulated genes in K9A (*n* = 1084) were related to immune response, extracellular matrix organization, and differentiation. Upregulated genes in K27A (*n* = 183) and in both K9A and K27A mutants (*n* = 377) were commonly enriched for development and differentiation processes, such as regionalization, pattern specification processes, and embryonic organ development. Downregulated genes

in K9A were related to protein secretion processes (*n* = 525) and, for both mutants, meiosis (*n* = 547). Only 52 genes were differentially expressed in opposite directions in K9A and K27A, and these were enriched for protein removal processes. Thus, although H3.3 represents only ~20% of the total histone H3 pool in mESCs, these single amino-acid substitutions lead to extensive changes in mESC transcription, indicating that H3.3K9 and K27 are required for maintaining transcription in mESCs.

### H3.3K9A and K27A mESCs impaired differentiation

Timing of lineage-specific gene expression is critical during development, with chromatin modifiers and epigenetic modifications modulating cell fate transitions[23]. To understand the impact of H3.3K9A and H3.3K27A on developmental gene expression, we examined changes in marker gene transcription of embryonic lineages. Ectoderm and mesoderm marker genes were particularly affected in K9A and K27A mESCs, with lineage markers of trophectoderm (*Cdx1*, *Cdx2*), mesoderm (*Tbxt*, *Snai1*, and *Twist*) and cardiac-mesoderm (*Mef2c*, *Tbx1*, and *Fgf10*) showing significant upregulation (Fig. 2a).

We investigated whether the activation of such genes in mutant mESCs could affect in vitro differentiation. Upon differentiation, K9A mESCs formed significantly smaller embryoid bodies (EBs) (Fig. 2b), and upon subsequent induction to neuro-ectoderm, they generated fewer neuronal progenitor cells (NPCs), failed to differentiate into neurons, and formed a heterogeneous population of cells characterized by a non-neuronal-like morphology, compared to control mESCs (Fig. 2c, d). K27A cells formed a homogeneous population of neurons with regular morphology, unlike K9A cells. NPCs marker expression (*Hash1*, *Neurod1*, *Pax6*, and *Sox2*) at day 8 of differentiation was significantly affected only in the H3.3K9A mutant (Supplementary Fig. 3a). Immunofluorescence staining for neuronal markers (Map2 and beta-III tubulin) at day 12 of differentiation showed a marked decrease in K9A but not in K27A cells (Fig. 2d and Supplementary Fig. 3b).

Cardiac-mesoderm differentiation of K9A mESCs revealed a significantly reduced number of cells at the intermediate stage of the differentiation (Fig. 2e). The K9A cardiac-mesoderm EBs failed to progress further differentiation. Cardiac-mesoderm EBs derived from K27A mESCs also exhibited reduced cell numbers at day 4 of differentiation (Fig. 2e) but were able to commit to the mesoderm lineage. However, the number of actively contracting colonies gradually declined during differentiation (Fig. 2f), indicating that K27A cells are unable to maintain a functional cardiomyocyte differentiated state.

### Growth delay and cell death in H3.3K9A mESCs

The differentiation defects in H3.3K9A mESCs imply that factors that regulate differentiation have been perturbed. Monitoring proliferation over six days revealed decreased numbers of K9A mESCs, but not of K27A mESCs, compared to the control (Fig. 2g). This could be due either to reduced proliferation and/or increased cell death. Cell cycle analysis in controls and mutants revealed the same percentage of cells in G1, S, or G2/M phases (Fig. 2h and Supplementary Fig. 3c), indicating that proliferation was unaffected in K9A and K27A mESCs. Pluripotency marker gene expression was overall maintained, except for *Nanog* and *Sox2*, which were decreased in both K9A and K27A mESCs (Fig. 2i). However, annexin V staining revealed significantly increased cell death in K9A mESCs, but not in K27A mESCs (Fig. 2j and Supplementary Fig. 3d), which was uncoupled to the accumulation of lipid reactive oxygen species (Supplementary Fig. 3e). Altogether, these results indicate that increased cell death causes reduced cell numbers in K9A mESCs.

### Distal regulatory element activation in H3.3K9A mESCs

H3.3K9A and K27A mESCs are viable, able to self-renew, and display substantial changes in gene expression compared to the wildtype.

Therefore, using ChIP-seq, we explored the impact of H3.3 substitutions on the chromatin environment in mESCs. H3K9me3 ChIP-seq and differential analyses revealed that, among ~48300 H3K9me3 regions, 8685 were differentially abundant (DA) in K9A mESCs, while only 2 were changed in K27A mESCs (Fig. 3a and Supplementary Fig. 4a). 8203 out of 8685 DA-H3K9me3 regions in K9A mESCs showed a reduced signal, compared to the control and the majority of them were located within heterochromatin (Fig. 3a). Genomic annotation of the DA-H3K9me3 regions revealed that: 68% were located in distal intergenic regions, 18% were in introns, 3% were within 2 kb of a transcriptional start site (TSS), and the remainder were located in exons and 5'-/3'-UTRs (Supplementary Fig. 4b). ChIP-seq of H3K27ac in control and K9A mESCs revealed a significant increase of this activating mark in 13177 regions (Fig. 3b), mostly in regions annotated as heterochromatin (Fig. 3b),

overlapping with the H3K9me3 decreased regions (Supplementary Fig. 4b).

To better understand the de-repressed H3K9me3 regions, we employed k-means clustering to group the DA-H3K9me3 peaks (n = 8203) based on their H3K9me3 signal. This resulted in two distinct clusters: Cluster 1, which fully lost the H3K9me3 signal in the K9A mutant, and Cluster 2, where the signal was reduced yet still present. For comparison, we included 8500 randomly selected non-DA-H3K9me3 peaks in the analysis as a control set (Fig. 3c). Compared to the non-DA regions, both Cluster 1 and 2 displayed elevated H3.3 deposition, indicating that H3.3 is the primary contributor to the H3K9me3 signal in these regions. Cluster 1 regions, compared to Cluster 2, showed a higher basal level of H3K27ac in control mESCs, which was further increased in K9A mESCs. The H3K9ac signal was increased in K9A mESCs, despite the enrichment

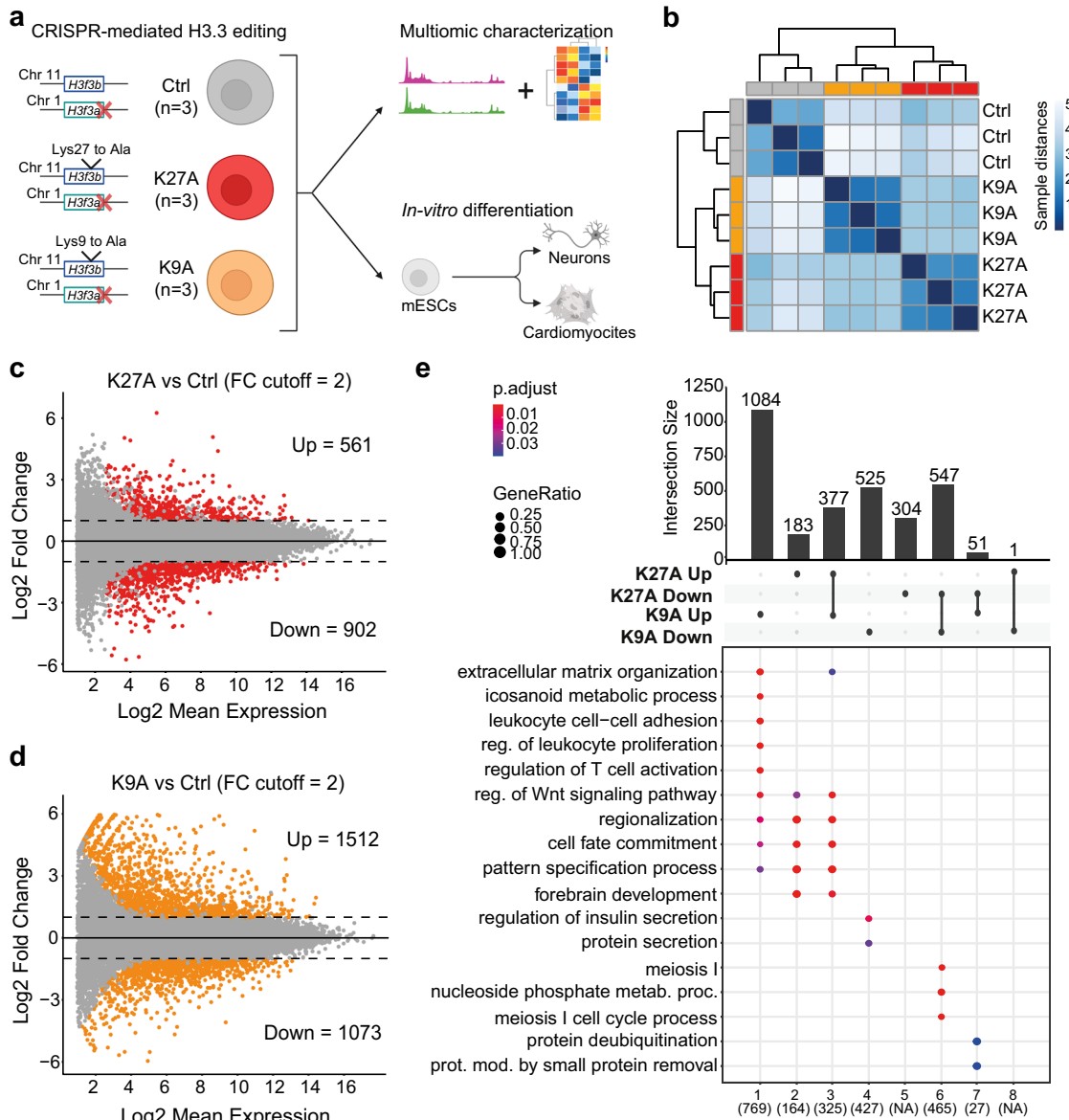

**Fig. 1 | Transcriptional changes in H3.3K27A and K9A mESCs. a** Graphical outline with mESCs lines genotype. **b** Hierarchical clustering with sample-to-sample distances computed from rlog-transformed mRNA-seq counts. **c** MA plots of DEGs in K27A versus control mESCs and **d** DEGs in K9A vs. control mESCs. Colored dots represent significant DEGs, defined with *p*adj <0.05 and abs(log2(fold-change)) cut-off = 1. *P*adj: Benjamini–Hochberg adjusted *p* values calculated with DESeq2. **e** Upset plot of DEGs up-/down-regulated exclusively in one mutant or shared between the two mutants. On the bottom, the most significant GO terms for each group of genes are reported. *P*.adjust: *p* values of significant GO terms adjusted for multiple comparisons. Figure 1a was created with BioRender.com and released under a Creative Commons Attribution-NonCommercial-NoDerivs 4.0 International license. Source data are provided as a Source Data file.

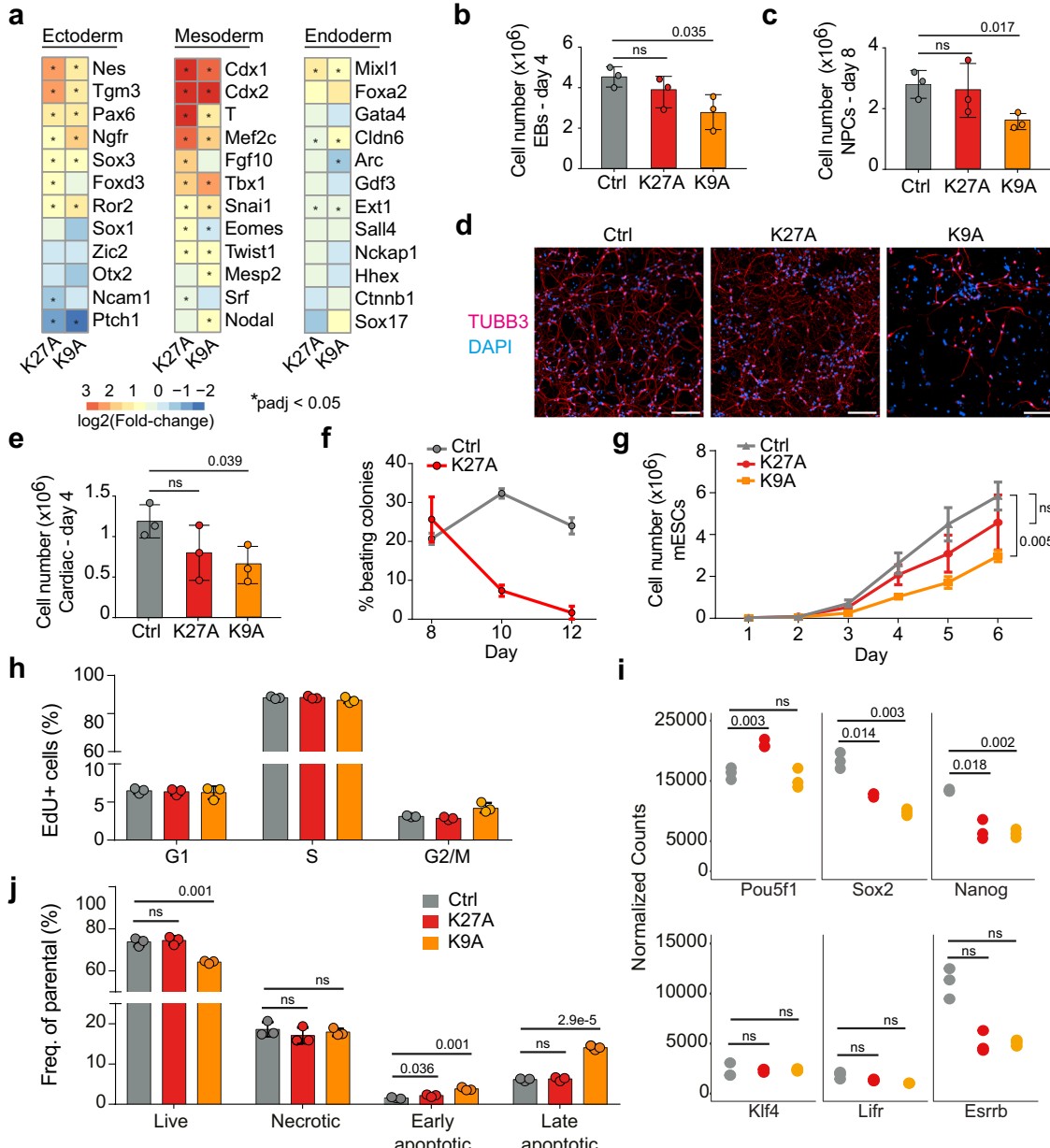

**Fig. 2 | H3.3K9A mESCs fail to differentiate and display increased cell death.**
**a** Heatmap of log2(fold-change) expression for selected ectoderm, mesoderm, and endoderm marker genes in K27A and K9A mESCs; asterisks indicate whether the gene is significantly differentially expressed in the mutant compared to the control. Significant differences were calculated with DESeq2 and adjusted with Benjamini−Hochberg's correction. **b** Barplot of cell numbers counted after dissociation of the embryoid bodies at day 4 and **c** at day 8 of the in vitro neuronal differentiation. **d** Merged immunofluorescence images of neurons on day 12 of the in vitro differentiation, stained with antibodies against TUBB3 to detect neurons and with DAPI to detect nuclei. Scale bar 100 μm. **e** Barplot of cell numbers counted after dissociation of embryoid bodies at day 4 of in vitro mesoderm-cardiac differentiation. **f** Line plot depicting the percentage of contracting colonies counted every 2 days throughout the final stage of the cardiac differentiation. **g** Growth curve for control and mutant mESCs. **h** Barplot with percentages of mESCs in G1, S, or G2/M phases detected following 5-ethynyl-2′-deoxyuridine (EdU) incorporation. **i** Normalized mRNA-seq counts for six selected pluripotency markers (Pou5f1/Oct4; Sox2; Nanog; Klf4; Lifr and Esrrb) in control, K27A and K9A mESCs. Significant differences were calculated with DESeq2 and adjusted with Benjamini−Hochberg's correction. (ns: *p*adj >0.05). **j** Barplot with percentages of cells identified as live, necrotic, early apoptotic, or late apoptotic after Annexin V staining. In panels **b**, **c**, **e**–**h**, **j**, data were mean ± standard deviation of *n* = 3 biological replicates. *P* value: two-sided unpaired *t*-test. Source data are provided as a Source Data file.

of mutant H3.3, indicating the presence of H3.1/H3.2 in the H3K9ac-enriched regions. Cluster 2 regions contained no basal histone H3 acetylation, regardless H3K9me3 levels were reduced, and there was a modest increase in H3K27ac and H3K9ac in K9A mESCs (Fig. 3c).

Given that Cluster 1 and 2 regions exhibited differences in H3K27ac upon H3.3K9A substitution, we used the ChIP-Atlas database[24] to compare the occupancy of chromatin modifiers and transcription factor (TF) binding between these two sets of regions in

wild-type mESC. Consistent with higher basal levels of H3K27ac, Cluster 1, compared to Cluster 2, showed a relative enrichment of transcriptional co-activators (P300, BRD4, and MED24), subunits of chromatin remodelers (SMARCA4, CHD7), and TFs linked to pluripotency (NANOG, SOX2, OCT4/POU5F1) and development (SOX17, SMAD2/4, and YAP1). Cluster 2 showed a relatively higher enrichment for core heterochromatin factors, such as HP1 proteins (CBX1, CBX3, CBX5), KAP1/TRIM28, SETDB1, and other transcriptional repressors (the STRING network of the factors shown in Fig. 3d).

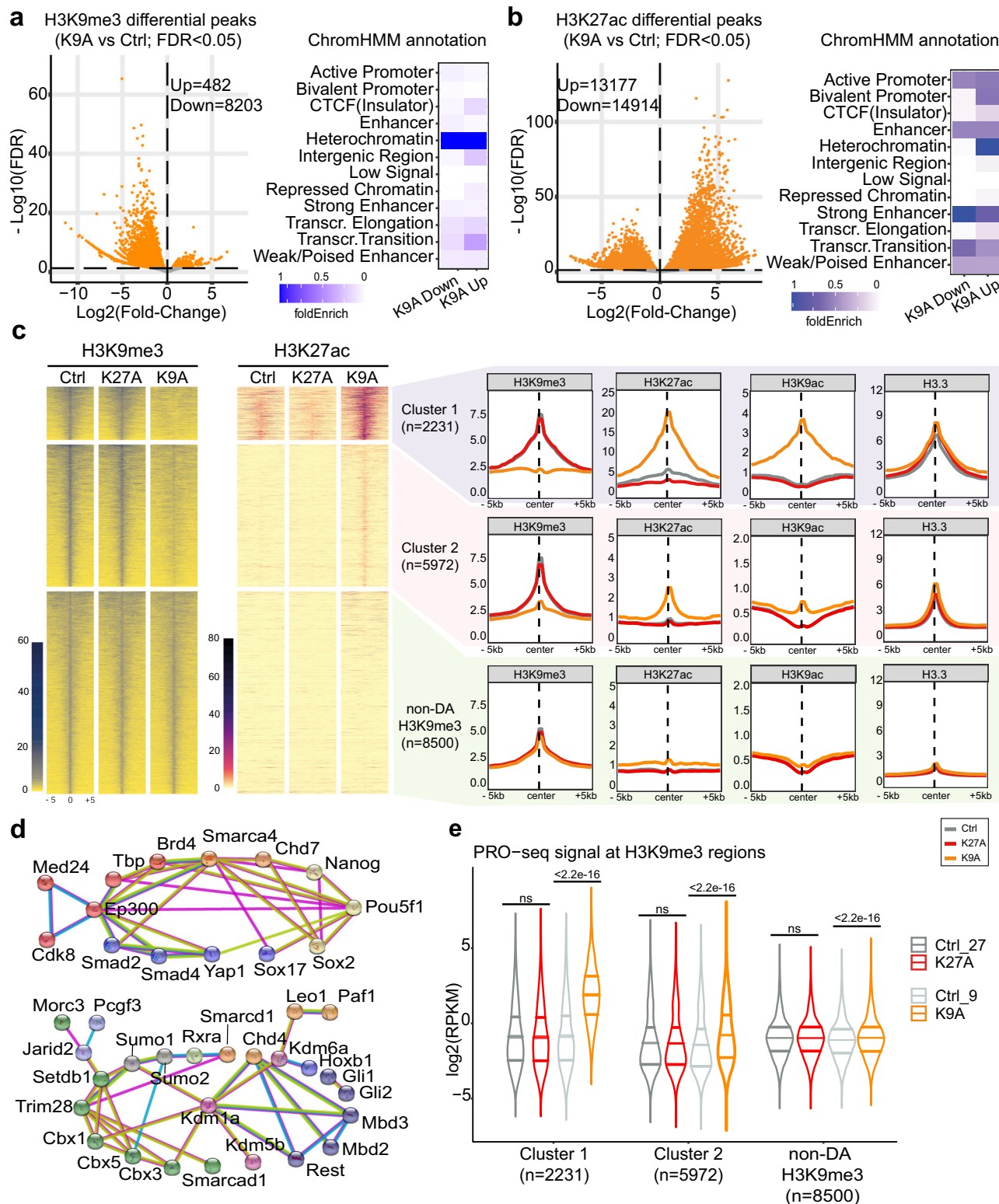

**a** H3K9me3 differential peaks (K9A vs Ctrl; FDR<0.05) — ChromHMM annotation

**b** H3K27ac differential peaks (K9A vs Ctrl; FDR<0.05) — ChromHMM annotation

**c** H3K9me3 / H3K27ac — Cluster 1 (n=2231), Cluster 2 (n=5972), non-DA H3K9me3 (n=8500)

**d**

**e** PRO−seq signal at H3K9me3 regions

To determine if these de-repressed H3K9me3 regions elicit transcriptional activity, we assessed nascent transcript levels (PRO-seq). Cluster 1 regions, which displayed a larger H3K9me3 loss and H3K27ac increase, showed a marked gain in nascent transcription (Fig. 3e). Cluster 2 regions showed a more modest increase in nascent transcription, consistent with the ChIP-seq data. Altogether, our results indicate that the H3.3K9 residue contributes to maintaining a set of distal intergenic regions in a repressed state and that the H3K9me3 signal reduction initiates a switch to an active chromatin state.

### De-repressed cryptic enhancers in H3.3K9A mESCs upregulate immune-related genes

We hypothesized that the de-repressed distal elements in Cluster 1 and 2 could act as cryptic enhancers, driving the expression of upregulated genes in K9A mESCs. To test this, we examined the expression of genes

**Fig. 3 | H3K9me3 reduction with a concurrent gain in H3K27ac and transcription at distinct heterochromatic regions in H3.3K9A mESCs.** **a** Volcano plot showing differential H3K9me3 ChIP-seq peaks in K9A vs. control mESCs (consensus peakset with $n = 48,346$ broad peaks) and ChromHMM annotation of DA-H3K9me3 regions. **b** Volcano plot showing differential H3K27ac ChIP-seq peaks in K9A vs. control mESCs (consensus peakset with $n = 94,000$ peaks) and ChromHMM annotation of DA-H3K27ac regions. **c** Heatmaps of H3K9me3 and H3K27ac ChIP-seq signals at H3K9me3 regions (summit ± 5 kb). Differentially abundant regions are divided into two clusters (k-means clustering) and sorted from a high-to-low H3K9me3 signal in control. ChIP-seq signal calculated as the average of three replicates for control or mutant mESCs, in 250 bp sliding genomic windows. On the right, metaprofile plots with H3K9me3, H3K27ac, H3K9ac, and H3.3 average ChIP-seq signals are reported. **d** STRING network analysis of chromatin and transcription factors enriched at Cluster 1 (top) and Cluster 2 (bottom) DA-H3K9me3 regions. Colors indicate the chromatin complex/category (red: transcriptional co-activator; orange: chromatin remodeler/elongation complex; yellow: pluripotency TF; blue: developmental TF; purple: histone demethylase; green: heterochromatin factor; dark-blue: transcriptional repressor; light-blue: Polycomb subunit). STRING settings: physical subnetwork (the edges indicate that the proteins are part of a physical complex) – medium confidence (interaction score 0.4). **e** Nascent transcription signal at DA-H3K9me3 regions reported as log2(RPKM) values. *P* value: two-sided unpaired *t*-test. PRO-seq experiments were performed independently for K27A and K9A. Control samples were repeated in each batch and reported in two gray shades. Source data are provided as a Source Data file.

within 50 kb of the DA-H3K9me3. Genes close to DA-H3K9me3 regions (especially Cluster 1 regions) were, on average, upregulated in H3.3K9A mESCs (Supplementary Fig. 4d). To further refine the putative enhancer-gene connections, we constructed an enhancer-mediated gene regulatory network (GRN) using GRaNIE[25], based on the co-variation between mRNA-seq and H3K27ac ChIP-seq data, from a total of 12 mESCs samples (see Methods). The GRN allows us to infer interactions between *cis*-regulatory elements (i.e., H3K27ac peaks) and genes. After filtering for significant connections, we obtained 16701 CRE-gene connections, composed of a network of 5229 genes and 5694 regulatory elements. In the GRN, a total of 657 DA-H3K9me3 regions were retained, which were mostly annotated as distal intergenic and intronic regions. These DA-H3K9me3 regions were linked to 789 genes ($n = 729$ in Cluster 1 and $n = 118$ in Cluster 2, with $n = 58$ genes that were linked to both clusters; average peak-gene distance ≈108 kb) and 82% ($n = 651/789$) of them were upregulated in K9A mESCs (Fig. 4a).

Many genes specifically upregulated in K9A mESCs were related to immune processes (Fig. 1e). We tested if the upregulation of immune-related genes could be driven by newly activated cryptic CREs. GO analysis of the genes connected to de-repressed CREs via the GRN revealed terms such as leukocyte-mediated cytotoxicity and regulation of response to a biotic stimulus (Fig. 4b). Individual genes underlying these terms included Toll-like receptor genes (*Tlr2*), class II major histocompatibility complex genes (MHC, e.g., *H2-M2* and *H2-T23*), interferon-inducible genes (*Mndal*), and members of the nucleotide-binding leucine-rich repeat gene family (*Nod1*) (Fig. 4b). Two additional GO terms related to inhibition of protease activities, including members of the serine protease inhibitors (serpins), and fatty acid and eicosanoid metabolism, both with reported roles in innate immunity[26,27], also showed enrichment among the upregulated genes linked to the de-repressed CREs in K9A mESCs. In addition, the increased transcription of immune genes in K9A mESCs led to protein expression changes, at least in three targets: CD59a (a surface glycoprotein involved in T-cell activation and inhibition of the complement membrane attack complex), TLR2 (a surface protein required for activation of innate immunity), and MHC Class II (surface proteins typically expressed in immune cells dedicated to the antigen presentation). Immunolabeling followed by flow cytometry analysis revealed significantly higher expression of these proteins in K9A mESCs compared to controls, while K27A mESCs showed no change (Fig. 4c).

To test whether the H3K9me3-marked cryptic CREs are active in immune cell types, we retrieved data from the mouse immune-system *cis*-regulatory atlas, which comprises over 500,000 putative CREs from 86 primary immune cell types, identified using ATAC-seq[28]. The comparison between the DA-H3K9me3 regions and the collection of immune open-chromatin regions (Imm-OCRs) revealed a significant amount of overlap in Cluster 1 (31%; $n = 697/2231$) and Cluster 2 (24%; $n = 1450/5972$), compared to the overlap in non-DA-H3K9me3 regions (14%; $n = 1247/8500$) (Fig. 4d). To determine if these CREs from Cluster 1 and 2 ($n = 2147$) are active in specific immune cells, we merged highly correlated Imm-OCRs from closely-related cell types and sorted the resulting cell types based on the activity (i.e., chromatin openness) of overlapping Imm-OCRs (Fig. 4e). Apart from stromal cells, the highest activity was observed in specialized immune cell types such as B-cells, T-cells, macrophages, and granulocytes. This result indicates that a significant fraction of the cryptic CREs de-repressed in H3.3K9A mESCs function as enhancers active in specialized immune cell types.

To examine whether immune-related TFs are active in H3.3K9A mESCs, we performed a differential TF activity analysis using diffTF[29]. By comparing K9A and control mESCs (*p*adj <0.001), we identified 84 differentially active TFs. HIVEP1/ZEP1 and HIVEP2/ZEP2, which bind viral promoters and CREs of immune-related genes[30,31], were amongst the TFs with the highest activity in K9A mESCs (Supplementary Fig. 5a). Other top hits included three NF-kB subunits (REL/C-REL; RELA/p65; NFKB1/p50), with roles in inflammation and in the immune response[32,33], and TAL1, a TF involved in hematopoiesis[34,35]. Given the activation of cryptic CREs associated with immune transcriptional programs and the increased immune cell surface markers in K9A mESCs, we differentiated control and K9A mESCs into macrophages in vitro[36], to explore potential functional effects. K9A cells exhibited premature expression of macrophage marker genes such as *Cd11b* and *Adgre1* (i.e., F4-80) at the early EB stage (day 8; Fig. 4f). However, the morphology of K9A EBs was irregular compared to controls (Supplementary Fig. 6a) and the premature expression of marker genes was not sustained in later differentiation stages (day 12; Fig. 4g). The resulting macrophage precursors derived from K9A cells were mostly non-viable (Fig. 4h) and failed to generate mature CD11b + /F4-80+ macrophages, as observed in control lines (Supplementary Fig. 6b).

To provide a direct relationship between H3.3K9-occupied cryptic CREs and the transcriptional activity of immune-related genes, we conducted rescue experiments involving the stable expression of histone H3 constructs (i.e., H3.3, H3.1, and H3.3K9A) in K9A mESCs. ChIP-qPCR analyses revealed an increased H3K9me3 signal at cryptic CREs linked to immune genes *Il7* and *Cd59a*, only in K9A cells expressing the H3.3 wild-type construct but not others (Fig. 4i). This led to reduced *Il7* and *Cd59a* expression specifically with H3.3 introduced cells as determined by RT-qPCR (Fig. 4j), along with decreased CD59a protein levels verified by immunolabeling and flow cytometry (Fig. 4k). Altogether, the de-repressed CREs in H3.3K9A mESCs regulate the expression of immune-related genes, culminating in the production of proteins restricted to specialized cell types. Nevertheless, these cells are unable to generate fully differentiated immune cells, likely due to the accumulation of regulatory defects in the genome.

## Cryptic enhancers in H3.3K9A mESCs derive from ERVs

Having identified H3.3-marked genomic regions acting as CREs upon the reduction of the H3K9me3 level, we sought to elucidate the unique features of these loci. Histone H3.3 is enriched at ERVs (also known as LTR elements)[37–39], which are a repertoire of genomic sequences with regulatory potential[40–43] and represent 15-30% of CREs in immune cells[44–47]. We investigated whether the cryptic CREs are derived from ERV sequences by assessing the degree of overlap between de-

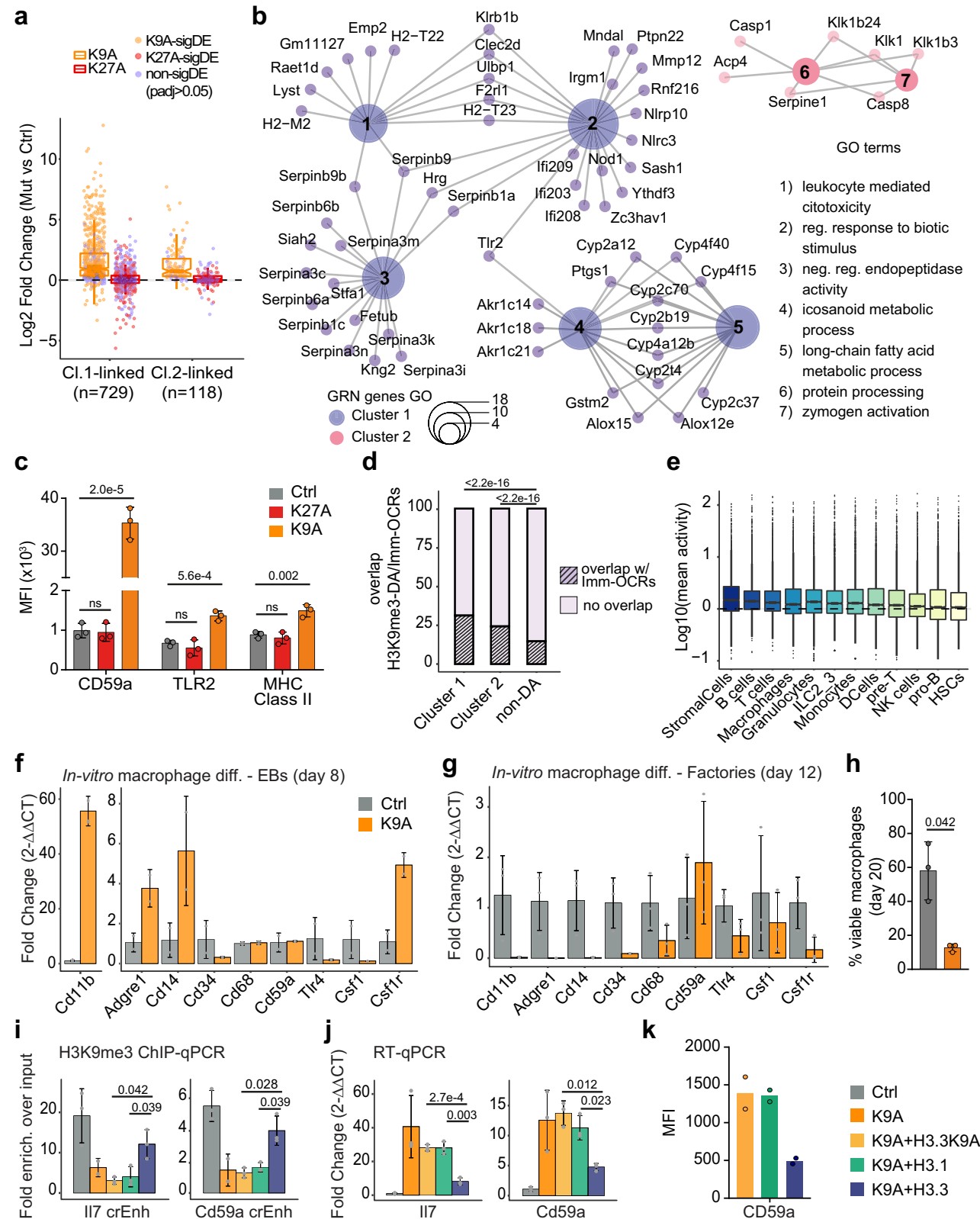

**c**

GO terms

1) leukocyte mediated citotoxicity
2) reg. response to biotic stimulus
3) neg. reg. endopeptidase activity
4) icosanoid metabolic process
5) long-chain fatty acid metabolic process
6) protein processing
7) zymogen activation

GRN genes GO

● Cluster 1
● Cluster 2

repressed CREs in K9A mESCs and ERVs. Analysis revealed substantial overlap, with 60% (n = 1347/2231) in Cluster 1 and 40% (n = 2322/5972) in Cluster 2, compared to 22% (n = 1916/8500) in the control set of non-DA-H3K9me3 regions (Fig. 5a).

To thoroughly examine transposable elements (TEs) activity in control and mutant cells, we conducted a transcriptomic analysis of repeat elements using the RepEnrich2 tool[48]. Control mESCs showed

higher basal expression of non-LTR TEs (Supplementary Fig. 7a) compared to LTR elements (mean log-transformed RNA normalized counts LINE = 2.76, SINE = 4.03, LTR = 2.26). Differential analysis of non-LTR TEs indicated modest changes in specific LINE and SINE subfamiliese (e.g., L1M3a, B2_Mm1a) in K9A mutants, but not in K27A mutants (Supplementary Fig. 7b). Regarding ERVs, control mESCs exhibited elevated expression of ERVK and MaLR subfamilies

**Fig. 4 | Cryptic CREs activated in H3.3K9A mESCs drive the expression of immune-related genes. a** Expression of genes linked to DA-H3K9me3 regions through the GRN; log2(fold-change) values calculated with DESeq2 are plotted, and individual genes (data points) are colored to indicate if the differential expression is significant (orange/red) or not (purple). **b** Cnet plot of gene ontology terms showing genes linked to DA-H3K9me3 regions in Cluster 1 (purple) and 2 (pink) through the GRN. **c** Barplots of median fluorescence intensity of selected surface markers. Data were mean ± standard deviation from $n = 3$ biological replicates of control (gray), K27A (red), or K9A (orange) mESCs. *P* value: two-sided unpaired *t*-test. **d** Stacked barplot displaying the percentage of overlap between DA-H3K9me3 regions and Imm-OCRs. Cluster 1, $n = 2231$; Cluster 2, $n = 5972$; non-DA, $n = 8500$. *P* value: one-sided two-proportions *Z*-test. **e** Mean activity of Imm-OCRs overlapping with DA-H3K9me3 regions in immune cell populations ($n = 3421$); cell types are sorted by decreasing mean activity of the open-chromatin regions. **f, g** RT-qPCR results for immune cell markers in embryo bodies (day 8; **f**) or factories (day 12; **g**) of macrophage differentiation in control and K9A cells ($n = 3$ independent clonal lines/condition). Data were the mean ± standard deviation of gene expression values ($2^{-\Delta\Delta Ct}$) normalized to Rpl13 housekeeping gene and control. **h** Percentages of viable macrophages at the terminal stage of macrophage differentiation (day 20), in control and K9A cells. Data were mean ± standard deviation from $n = 3$ independent clonal lines per condition. **i** H3K9me3 ChIP-qPCR at selected cryptic enhancer regions, connected through the GRN to Il7 and Cd59a genes, in control, K9A and K9A mESCs stably expressing H3.3K9A, H3.1 or H3.3WT. The relative enrichment over the input is reported and data were mean ± standard deviation ($n = 3$ independent replicates). crEnh cryptic enhancer. **j** qRT-PCR results for Il7 and Cd59a genes in control, K9A and K9A cells stably expressing H3.3K9A, H3.1 or H3.3WT. Data were the mean ± standard deviation of gene expression values ($2^{-\Delta\Delta Ct}$) normalized to Rpl13 housekeeping gene and control ($n = 3$ independent replicates). *P* value: two-sided unpaired *t*-test. **k** Barplots of median fluorescence intensity of CD59a surface marker ($n = 2$ biological replicates of K9A mESCs stably expressing H3.3K9A, H3.1, or H3.3WT). Source data are provided as a Source Data file.

(Supplementary Fig. 7c), consistent with previous findings[40,43,49,50]. For instance, RLTR9s—acting as stem cell-specific regulatory elements[43,50]—displayed high expression levels. Among ERV families (ERVK, ERV1, ERVL, and MaLR), the ERVK showed significantly increased transcription in K9A mESCs (Supplementary Fig. 7c). Differential ERVs expression analysis identified MMERVK, RLTR, IAP, and IAPE elements as significantly upregulated in K9A mutants compared to controls (Fig. 5b and Supplementary Fig. 7d, e). RT-qPCR validated increased transcriptional activity of IAP in K9A mutants (Supplementary Fig. 7f), with decreased expression of IAP gag observed in rescued lines expressing the H3.3 wild-type construct (Fig. 5c).

ChIP-seq analysis at repeat elements using the T3E tool[51] revealed significant H3.3 enrichment over input at ERVs (35%; $n = 236/659$, *p* val <0.01), particularly at ERVK and ERV1 (Fig. 5d), and at SINE (83.7%; $n = 31/37$)—although with lower overall enrichment over input compared to LTR elements (mean log2FC = 1.05 for LTRs, 0.46 for SINE; Supplementary Fig. 7g, h). Further ChIP-seq analyses focusing on ERVs with higher H3.3 enrichment ($n = 108$; log2 Fold-change cut-off = 1) revealed globally unaffected H3.3 occupancy in the mutants (Fig. 5e, h). However, H3K9me3 levels were significantly reduced in K9A mutants, accompanied by increased H3K27ac levels (Fig. 5f–h). Independent H3K9me3 ChIP-qPCR confirmed signal reduction at ERVK family members, such as IAP sequences in K9A mESCs (Supplementary Fig. 7i), with restoration observed in rescue experiments expressing the H3.3 wild-type construct, but not the H3.3K9A or H3.1 constructs (Fig. 5i), suggesting a causal role of H3.3K9.

Regarding ERVs active in K9A mutants, identified by both H3K27ac and transcription, they can be classified into three groups: (i) ERVs marked by H3K27ac and low levels of H3K9me3, already active in control mESCs and maintained in K9A mESCs (RLTR17, RLTR13E, RLTR9D, ERVB2_1A-I_MM-int, IAPEY4_LTR, etc.) (Supplementary Fig. 7j). (ii) ERVs displaying H3K9me3 and H3K27ac signal in control mESCs, which showed increased activation in K9A mutants (BGLII, MMERVK9C_I-int, ETnERV-int, LTRIS2, LTRIS3, RLTR1B, RLTR13G, etc.). (iii) ERVs lacking H3K27ac signal in control mESCs, but specifically activated in K9A mESCs (IAPA_MM-int, IAPLTR2_Mm, LTRIS-4/5, LTRIS_Mus, RLTR1, RLTR10-B/C/D, RLTR27, etc.) (Fig. 5h). Considering ERVs as cryptic *cis*-regulatory elements, we examined TF motifs, revealing enrichment for pluripotency-related (POU5F1, SOX2/SOX9, ETS family members) and inflammation-related TF motifs (IRF, NF-kB, MAFK, MITF) in ERVs with enhanced activation in K9A mutants. Development-related (HOX, ZBTB, and TBX families) and immune/inflammation-related (MAFK, REL/RELA, SPIC) TF motifs were enriched in ERVs exclusively activated in K9A mESCs (Fig. S7k-l), confirming their putative regulatory role.

While ERVs function as regulatory elements, they also trigger an interferon-mediated antiviral response, known as viral mimicry[52,53]. However, viral infection or exposure to synthetic viral RNA does not induce significant interferon responses in mESCs due to low expression levels of major innate immunity factors[54,55]. We did not observe a global upregulation of interferon-stimulated genes in K9A mESCs (Supplementary Fig. 8a). Treatment with RIG012[56], an inhibitor of the RIG-I receptor responsible for detecting cytosolic viral RNA[57,58], maintained increased expression of immune genes in K9A mESCs, comparable to the activation observed with the DMSO control (Supplementary Fig. 8b). This indicates that viral mimicry is not the primary driver of immune gene activation in K9A cells. Together, these findings suggest that the vast majority of activated cryptic CREs in H3.3K9A mESCs are derived from ERVs, some of which can drive the expression of immune-response genes.

## Enhancer rewiring in H3.3K9A mESCs

Inspection of Cluster 1 and 2 regions confirmed ERV presence within regions linked through the GRN to immune-related genes that were upregulated in K9A mESCs, such as *Il7*, *Fndc4* (Fig. 6a, b), *H2-Bl*, and *Cd59a* (Supplementary Fig. 9a, b).

The de-repression of numerous ERV-derived CREs in K9A mESCs could induce changes in the distribution of transcription factors and chromatin components involved in gene regulation. Analysis of the PRO-seq data with the dREG and tfTarget packages[59–61] revealed 98 and 20 TF motifs significantly enriched at CREs with increased and reduced activity in K9A mESCs, respectively. TF motifs at more active CREs were related to immune cell specification (BHLHE40, TCF3 and IRFs), NF-kB pathway, developmental processes (TBX2, TBX3, SOX9, SOX17, HOXA3 and HOXD3), and pluripotency maintenance (POU5F1/OCT4, SOX2 and MYC). TF motifs at CREs with reduced activity were related to cellular function (FOS and JUN) and pluripotency (POU5F1/OCT4 and ESRRB) (Supplementary Fig. 10a, b), suggesting a decrease in the activity of pre-existing canonical enhancers. Consistently, there was a decrease in 14,914 H3K27ac peaks in K9A mESCs alongside an increase in 13,177 peaks (Fig. 3b). Furthermore, within the Cluster 1 regions, TF motifs crucial for pluripotency maintenance, such as POU5F1, SOX2/SOX9, and the OCT4::SOX2::TCF::NANOG (OSTN) consensus motif (Fig. 6c) were enriched, mirroring TF motifs found at canonical enhancers (Fig. 6d).

We investigated whether the activity of canonical enhancers is altered in K9A mESCs concomitantly with the activation of cryptic enhancers and whether this reflects changes in gene expression. Among the active canonical enhancers identified using dREG, 934 were linked to at least one gene in the GRN. We divided these into three groups based on their nascent transcript reads in control mESCs ($n = 568$ strong; $n = 309$ medium; $n = 57$ weak). In K9A mESCs, nascent transcription levels were mainly reduced at strong enhancers, medium enhancers did not show a significant decrease, and weak enhancers showed increased activity (Fig. 6e), even though the H3K27ac signal was significantly reduced across all three groups (Supplementary

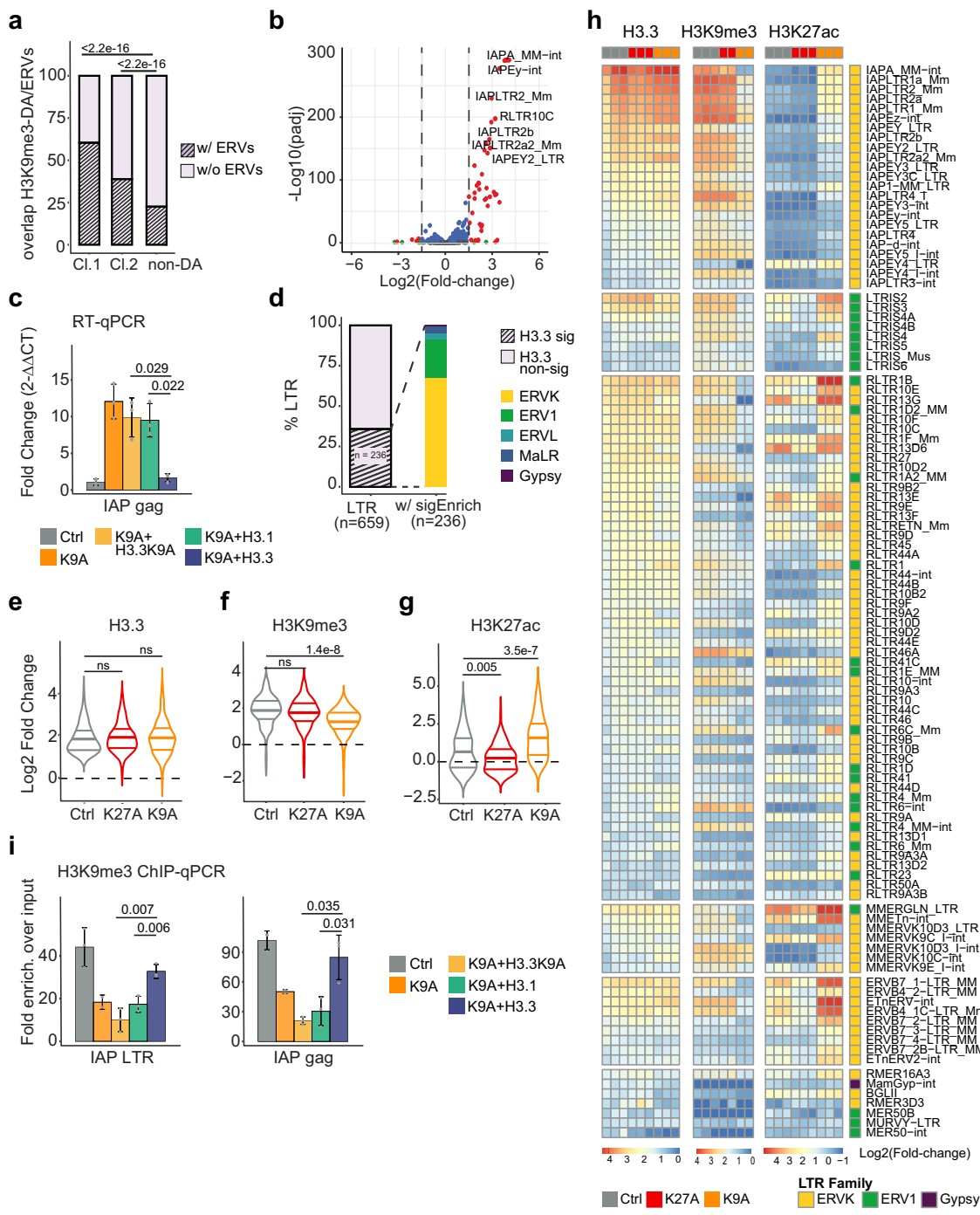

**Fig. 5 | ERVs de-repression in H3.3K9A mESCs. a** Stacked barplot showing the percentage of DA-H3K9me3 regions with at least one ERV. *P* value: one-sided two-proportions *Z*-test. **b** Volcano plot showing differential ERVs expression (*p*adj <0.05 & abs(log2FC) >1.5) in K9A vs. control mESCs. *P*adj: Benjamini–Hochberg adjusted *p* values calculated with DESeq2. **c** RT-qPCR results for IAP gag sequence in control, K9A and K9A cells stably expressing H3.3K9A, H3.1, or H3.3WT. Data were the mean ± standard deviation of gene expression values ($2^{-\Delta\Delta Ct}$) normalized to Rpl13 housekeeping gene and control (*n* = 3 independent replicates). *P* value: two-sided unpaired *t*-test. **d** Stacked barplot indicating the percentage of ERVs with significant (*p*val <0.01) H3.3 enrichment (left), and the classification of ERV families with significant H3.3 signal (right). *P*val: empirical *p* value calculated with T3E. **e**–**g** Violin plots displaying **e** H3.3, **f** H3K9me3, **g** H3K27ac log2(Fold-change) signal at ERV

subfamilies (*n* = 108), in control, K27A and K9A mESCs. *P* value: two-sided unpaired *t*-test. **h** Heatmaps displaying H3.3, H3K9me3, and H3K27ac log2(fold-change) signal at ERV subfamilies with highest H3.3 signal (*p* val <0.01 & log2FC >1; *n* = 108). ERV subfamilies are divided into six groups, and sorted in each group by decreasing H3.3 signal in control mESCs. For each subfamily, the corresponding ERV family is reported and color-coded (ERVK in yellow; ERV1 in green, and Gypsy in purple). *P* value: empirical *p* value calculated with T3E. **i** H3K9me3 ChIP-qPCR at ERV sequences in control, K9A and K9A mESCs stably expressing H3.3K9A, H3.1, or H3.3WT. The relative enrichment over the input is reported and data were mean ± standard deviation (*n* = 3 independent replicates). *P* value: two-sided unpaired *t*-test. Source data are provided as a Source Data file.

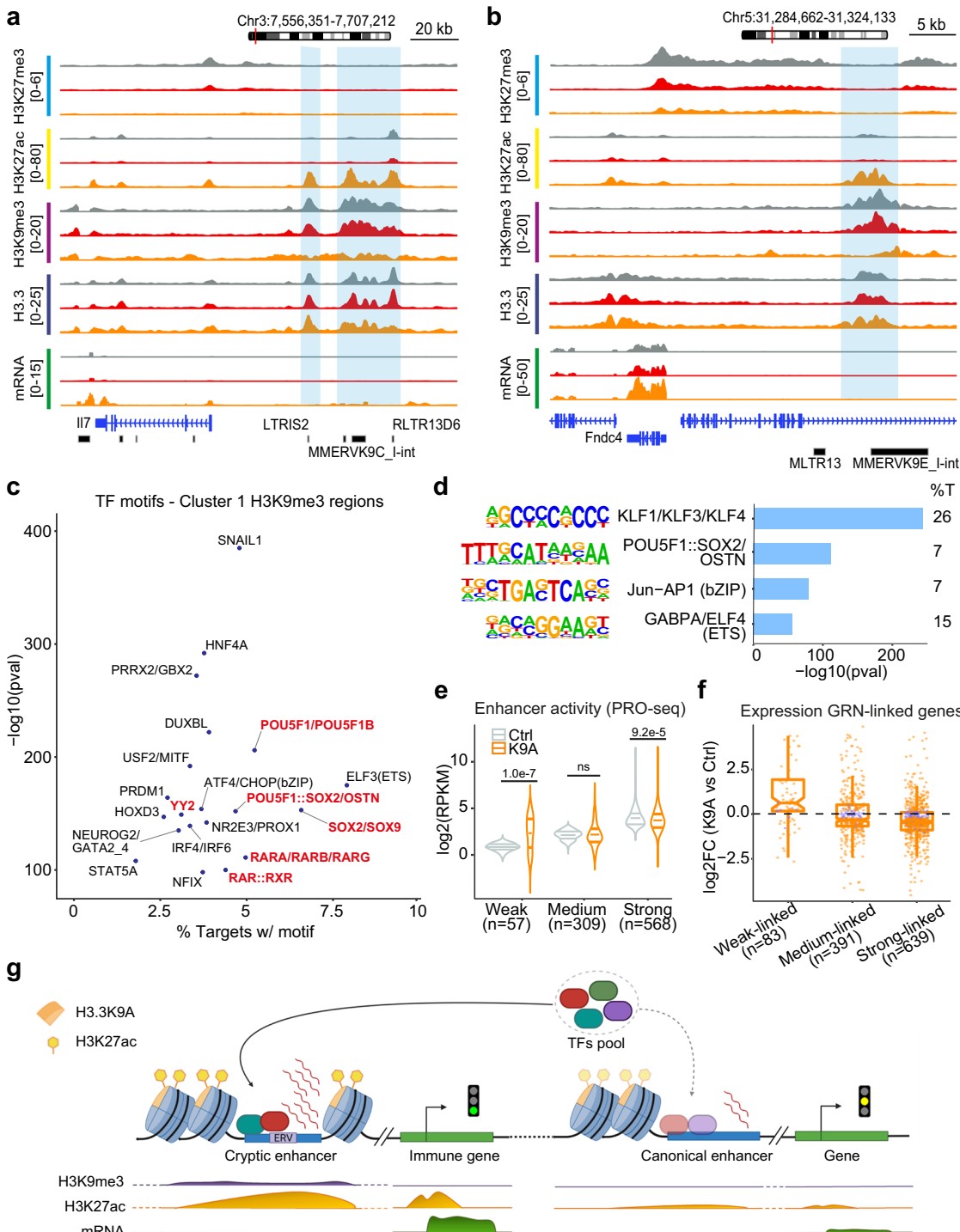

**Fig. 6 | Enhancers rewiring in H3.3K9A mESCs. a, b** Genome browser snapshots of representative Cluster 1 DA-H3K9me3 regions (highlighted), linked through the GRN to the Il7 (**a**) and Fndc4 (**b**) genes; H3K27me3, H3K27ac, H3K9me3, H3.3, and mRNA-seq tracks are shown. **c** Scatter plot displaying the top 20 transcription factor motifs significantly enriched at Cluster 1 DA-H3K9me3 regions. Pluripotency-related TF motifs are indicated in red. **d** Top 4 TF motifs significantly enriched at Strong dREG enhancers. %T: percentage of targets with motif. *P* values are calculated with Homer. **e** PRO-seq signal at Weak, Medium, and Strong dREG enhancers included in the GRN for K9A and control mESCs. Log2(RPKM) values are reported. *P* value: two-sided unpaired *t*-test. **f** Expression of DEGs connected to dREG enhancers through the GRN; log2(fold-change) values calculated with DESeq2 are plotted, and individual genes (data points) are colored to indicate if the differential expression is significant (orange) or not (purple). **g** Model of *cis*-regulatory elements rewiring in H3.3K9A mESCs. Figure 6g was created with BioRender.com released under a Creative Commons Attribution-NonCommercial-NoDerivs 4.0 International license. Source data are provided as a Source Data file.

Fig. 10c). We identified 639, 391, and 83 DEGs connected to strong, medium, and weak enhancers, respectively (average peak-gene distance ≈111 kb). Consistent with the reduction in enhancer activity, genes connected to strong enhancers were, on average, downregulated (Fig. 6f). Genes connected to medium and weak enhancers either showed a trend towards downregulation or significant upregulation, respectively, correlating with the differences in enhancer activity.

As a comparison, given that H3K27ac showed a marked decrease also in K27A mESCs (Supplementary Fig. 10d), we analyzed canonical enhancer activity and associated gene expression in K27A mESCs. Nascent transcription levels were significantly reduced at both strong and medium enhancers (Supplementary Fig. 10f), correlating with overall reduced expression of connected genes (Supplementary Fig. 10g). The correlation between enhancer activity and associated gene expression was not observed in a recent study with H3.3K27R mESCs[17], implying a stronger effect of the H3.3K27A substitution at CREs. Indeed, assessing H3K9ac and H3K18ac signals revealed their decreases in K27A mESCs (Supplementary Fig. 10e, h), which was not seen in K27R mESCs[17] nor in K9A mESCs.

Taken together, these analyses indicate that substantial changes in the chromatin state of H3.3K9me3-marked CREs reshape gene regulatory networks in K9A mESCs, with de-repressed ERV-derived CREs upregulating a subset of genes in K9A mESCs, and potentially providing a plethora of TFs binding sites[40,62], resulting in a redistribution of TFs and subsequent gene expression changes in K9A mESCs (Fig. 6g).

## H3K27me3 reduction at bivalent promoters in H3.3K9A mESCs

Having detailed immune-response genes upregulated in K9A mESCs, we studied genes associated with developmental processes, many of which were upregulated in both K9A and K27A mESCs (Fig. 1e). We focused on the promoter region of these developmental genes, which are typically enriched with repressive H3K27me3, forming bivalent promoters[63]. We assessed the H3K27me3 abundance at bivalent promoters in control and mutant mESCs using ChIP-seq. Consistent with H3.3 enrichment at bivalent promoters ($n = 4117$; H3K27me3 + H3K4me3, see Methods) (Fig. 7a), the H3K27me3 signal at these regions was reduced in K27A mESCs (Fig. 7b), accompanied by a modest increase in H3K27ac (Fig. 7c and Supplementary Fig. 10d). Additionally, we observed a decrease in the H3K27me3-signal at bivalent promoters in K9A mESCs. Histone mass-spectrometry data also revealed a global reduction in H3K27me3 levels in K9A mESCs (Supplementary Fig. 2).

We investigated the role of the H3.3K9 residue in the proper deposition of H3K27me3 at bivalent promoters. As the PRC2 complex deposits H3K27me3, we mapped the occupancy of the PRC2 core subunit SUZ12 in control and mutant mESCs. SUZ12 binding was more substantially affected in K27A (1062 SUZ12 peaks showed a decrease) than K9A mESCs (41 peaks decreased: Fig. 7d and Supplementary Fig. 11a), indicating a minimal effect of H3.3K9 residue on PRC2 occupancy. Allosteric activation of PRC2 is required for the proper regulation of H3K27me3 deposition and spreading[64,65]. We assessed the impact of the H3.3K9A and K27A mutations on this process by analyzing H3K27me3 distribution over SUZ12 peaks[66]. While H3K27me3 signals within SUZ12 peaks were similarly reduced in both K27A and K9A mutants compared to control, H3K27me3 levels were further decreased in the immediate proximity of the SUZ12-bound regions in K27A mESCs only (Fig. 7e), indicating that the H3.3K9 residue does not affect PRC2 allosteric activation.

Prior studies reported a putative function for H3K9 methylation in PRC2 activity, in which G9a and SETDB1 (H3K9 di- and tri- methyltransferases, respectively) reinforce PRC2 activity at target promoters[67–69]. ChIP-seq for H3K9me2 and H3K9me3 revealed that these marks were scarce at bivalent promoters (Fig. 7f). PRC2 activity assays using reconstituted chromatin composed of the *Atoh1* bivalent locus DNA sequence combined with H3.1 or H3.3 histone variants marked with H3K9me2 or H3K9me3 did not show enhanced PRC2 activity (H3K27me3 increase; Fig. 7g). However, we found that chromatin reconstituted with un-methylated H3.3K9 was a better PRC2 substrate than H3.1K9 or di−/tri−methylated H3.3/H3.1K9 (Fig. 7g). This result suggests that the H3.3K9 residue at Polycomb target loci could directly enhance the H3K27me3 signal, explaining the overall reduced H3K27me3 signal at PRC2-target promoters in K9A mESCs.

## De-repressed cryptic enhancers in H3.3K9A mESCs upregulate bivalent genes

Although the H3K27me3 signal was reduced at bivalent promoters, only a subset of bivalent genes showed a significant increase (fold-change cut-off of 2) in mutant mESCs. Bivalent genes upregulated in K27A and shared between the mutants ($n = 323$) were enriched with typical developmental GO terms, whereas those upregulated only in K9A mESCs ($n = 340$) showed divergent developmental GO terms such as synaptic transmission and leukocyte differentiation (Supplementary Fig. 11c). We compared basal H3.3 occupancy at promoter regions of these genes using control mESCs. Bivalent genes upregulated in both mutants had significantly higher basal levels of H3.3 than those upregulated only in the K9A mutant or of randomly selected non-differential bivalent genes (Supplementary Fig. 11d). This indicates that bivalent promoters with higher basal levels of H3.3 were more susceptible to the decrease in H3K27me3 upon H3.3K27A and H3.3K9A substitutions (i.e., H3.3K9 increases H3K27me3) and more likely to be de-repressed, suggesting direct causation. The cell lineage marker genes upregulated in both K27A and K9A mESCs (Fig. 2a and Supplementary Fig. 11c) are examples of these bivalent promoter-containing genes.

We investigated whether the increased expression of bivalent genes exclusive to K9A mESCs resulted from the activation of distal regulatory elements. Among the genes included in the GRN, we identified 153 out of 789 DA-H3K9me3-linked genes with a bivalent promoter signature (Fig. 7h). Of those, 68 bivalent genes were exclusively upregulated in K9A mESCs, indicating that the de-repressed cryptic CREs could activate this subset of bivalent genes. Some of these genes, such as *Tbx20* (Fig. 7i), *Frzb* (Supplementary Fig. 11e) and *Chrd* (Supplementary Fig. 9c), were involved in developmental processes and contained ERVs in the de-repressed CREs. Upon expression of the H3.3 wild-type construct in K9A mESCs, we observed increased H3K9me3 signal at cryptic CREs linked to the *Tbx20* and *Frzb* genes, and a concomitant reduced expression of the two genes (Supplementary Fig. 11f, g). Thus, we conclude that the upregulation of certain developmental genes in K9A mESCs is driven by the de-repression of cryptic ERV-derived CREs, similar to genes related to immune responses.

## Discussion

Here, our comparative systemic analysis demonstrates that H3.3K9 and H3.3K27 mediated PTMs are crucial for orchestrating repressive chromatin states at *cis*-regulatory elements and bivalent promoters, respectively, and for instructing proper transcription in mESCs. We have detailed how the removal of the K9 and K27 residues of histone H3.3 perturbs the histone modification landscapes in mESCs, and how those chromatin changes interplay to regulate gene expression.

Acetylation and methylation of H3K9 and H3K27 are involved in gene activation and repression, respectively. We found that changes in H3K9ac and H3K27ac in mutant mESCs merely correspond with enhancer activity. Our study revealed that a reduction in the level of H3K9me3 in H3.3K9A mESCs and H3K27me3 in H3.3K27A mESCs directly contributes to the gene activation observed in the individual mutants. In K9A mESCs, a decrease in H3K9me3 at numerous heterochromatic regions resulted in the activation of cryptic enhancers. Some of them have a distinctive genomic feature (i.e., Cluster 1) marked by H3K9me3, yet (i) harboring H3K27ac and (ii) having a higher enrichment for binding of TFs and transcriptional co-activators, compared to other heterochromatic regions in WT mESCs. These unique distal genomic regions, displaying high H3.3 occupancy, are therefore primed to be activated. We showed how these regions are strictly controlled by H3.3K9me3, and the reduction of this mark is sufficient to trigger a switch to an active chromatin state. By inferring

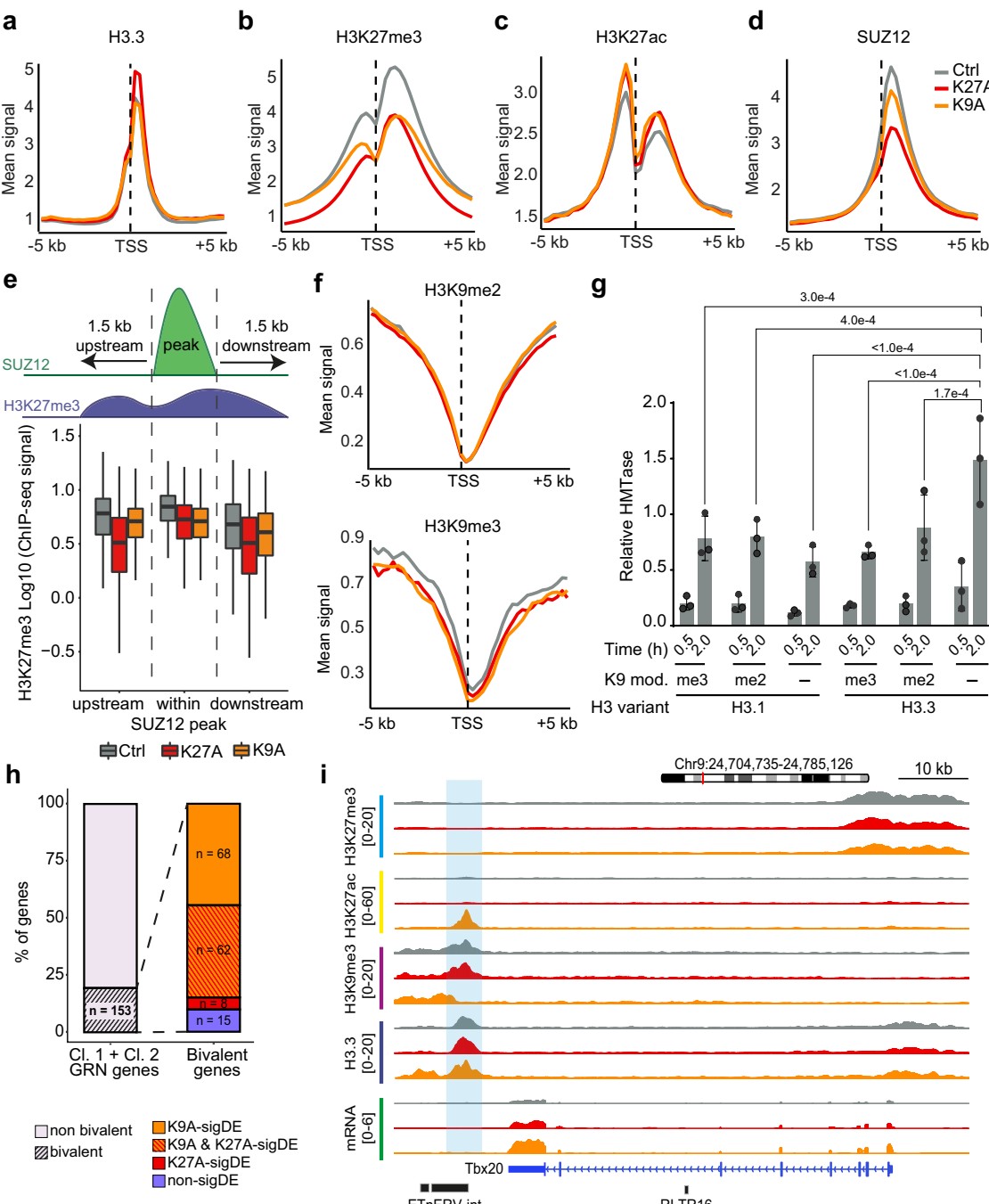

**Fig. 7 | H3.3K9A affected H3K27me3 level at bivalent promoters and increased bivalent genes via de-repressed cryptic enhancers. a–d** Metaprofile plot of **a** H3.3, **b** H3K27me3, **c** H3K27ac, and **d** Suz12 ChIP-seq signal at TSS ± 5 kb of bivalent genes (*n* = 4117). **e** Analysis of H3K27me3 signal within SUZ12 peaks (*n* = 3356) and in the 1.5 kb genomic region immediately upstream or downstream. **f** Metaprofile plot of H3K9me2 (left) and H3K9me3 (right) ChIP-seq signal at TSS ± 5 kb of bivalent genes (*n* = 4117). **g** Histone methyltransferase assays of PRC2 in the presence of chromatin reconstituted using different H3 variants and H3K9 modifications. In all cases, "me2" and "me3" represent the respective methyl lysine analog (MLA), and "-" represents an unmodified lysine residue. The barplot represents the mean densitometry values as recorded from the radiograms (see

Supplementary Fig. 11b and Source Data file). The error bars represent the standard deviation over three replicates carried out on different days. *P* value: Tukey's multiple comparisons test. **h** Stacked barplot indicating the percentage of genes with or without bivalent promoters (left), and details of significant differential expression (right). **i** Genome browser snapshot of a representative Cluster 2 DA-H3K9me3 region (highlighted), linked through the GRN to the Tbx20 gene; H3K27me3, H3K27ac, H3K9me3, H3.3, and mRNA-seq tracks are shown. For metaprofiles in panels **a–d**, **f**, the ChIP-seq signal is calculated as the average of three biological replicates for control or mutant mESCs, in 250 bp sliding genomic windows. Source data are provided as a Source Data file.

---

enhancer-gene connections with the GRN generated in this study, we could identify genes presumably regulated by these cryptic CREs, such as those involved in immune and developmental processes.

H3.3 has been associated with heterochromatin organization due to increased activation of heterochromatin elements, such as ERVs,

observed in H3.3 knockout mESCs or H3.3 chaperone knockout mESCs[37,70,71]. However, the precise contribution of H3.3, specifically H3.3K9, to heterochromatin function remains unclear[70–72]. We investigated its impact on chromatin landscape and transcription by introducing the K9A substitution in the endogenous H3.3. We found

that the removal of the H3.3K9 residue led to a decrease in the H3K9me3 signal at specific genomic regions (about 18% of total H3K9me3 domains), particularly at ERVK (IAPs, IAPLTRs, and IAPEs) and ERV1 (RLTR1s), aligning with the result in H3.3 knockout mESCs[37]. Rescue experiments confirmed the central role of H3.3K9 in H3K9me3 accumulation and transcriptional repression, particularly at IAPs.

Furthermore, we demonstrated the essential role of H3.3K9 in suppressing gene regulatory activities associated with specific ERVs, leading to abnormal activation of transcriptional programs in K9A mESCs. Compared to H3.3 knockout mESCs, H3.3K9A mESCs displayed more prominent transcriptional dysregulation, likely due to the preservation of histone variant incorporation into chromatin. Additionally, unlike KAP1/TRIM28 knockouts, which induce significant ERV activation[73,74] but exhibit a rapid decline in mESCs viability[41,75], H3.3K9A mESCs can be maintained in culture, facilitating the comprehensive investigation of the transcriptional rewiring upon ERV activation. Thus, the H3.3K9A approach offers unique insights distinct from the H3.3 knockout or H3.3 chaperone knockout methods.

Nevertheless, our study does not define the chaperone system responsible for the H3.3K9A mutant deposition. H3.3 is associated with both heterochromatin and active transcription, requiring histone turnover. The DAXX/ATRX/MORC3 chaperone pathway deposits H3.3 in heterochromatin, while HIRA deposits H3.3 in euchromatin/sites of active transcription[76]. In the DAXX-related pathway, H3.3 can be pre-modified with K9me3 before deposition[77]. The H3.3K9A mutant appears compatible with heterochromatin deposition, but whether DAXX or HIRA mediates this process is unclear. Transcriptionally active ERVs may have switched to HIRA deposition due to their transcriptional activity and changes in chromatin state.

We demonstrated that the activated cryptic CREs in H3.3K9A mESCs originate from de-repressed ERV sequences. Previous studies reported the expression of specific ERV subfamilies in mouse and human pluripotent stem cells[40,43,49,78], particularly ERVK and MaLR families in mESCs, which harbor pluripotency TF binding sites and act as CREs involved in early embryonic gene expression program[43,49]. Some ERVs in mESCs have been shown to display both repressive H3K9me3 and active H3K27ac histone marks[74]. In alignment with this, our study identified three distinct groups of ERVs based on their initial activity level and response to the H3.3K9A mutation: those marked by H3K27ac in control mESCs and similarly active in K9A mutants, those exhibiting H3K9me3 and H3K27ac marks in control mESCs and further activated in K9A mutants, and those typically inactive with H3K9me3 mark in control mESCs but activated in K9A mutants. The activity of these ERVs in mESCs appears to be controlled by a delicate interplay between repressive and activating chromatin signals. Disrupting this balance in H3.3K9A mESCs led to excessive activation of ERVs, highlighting the critical role of H3.3K9 in governing ERVs and varying their baseline activity levels in mESCs. Future studies investigating the function of H3.3K9-mediated ERV dynamics beyond mESCs will be a significant area of research.

Transposable element sequences contain binding sites for several TFs[40,42,79,80]. The activation of ERV-derived CREs in H3.3K9A mESCs potentially creates competition with most active canonical enhancers (namely "strong"), by making TFs binding sites accessible in heterochromatic territories[62] and thus acting as sponges. Therefore, changes in H3K27ac occur indirectly in H3.3K9A mESCs and are overall correlated with rewired enhancer activities.

On the contrary, deposition of the H3K27ac mark is directly affected at canonical enhancers in H3.3K27A mESCs, and the H3K27ac decrease was accompanied by reduced acetylation at other H3 lysines. Nascent transcription levels were reduced, and the genes connected to these regions through the GRN were downregulated, suggesting that the overall level—rather than residue-specific—H3 acetylation may play a role in maintaining enhancers in an active state.

The modification status of histone residues can affect the deposition of other target residues[2,81]. Here, our biochemical assay showed that unmodified H3.3K9 is a better substrate for PRC2 than H3.1K9, and thus the presence of H3.3K9A at Polycomb target promoters may have a direct negative impact on PRC2 activity, resulting in decreased H3K27me3 levels.

Regarding the H3.3K27A effect on H3K27me3, consistent with recent studies in Drosophila[82] and mammals[17,83–86], the H3.3K27 residue contributes to properly establishing the H3K27me3 mark at PRC2-bound promoters, as well as to the spreading of the modification. A decrease in promoter H3K27me3 levels does not always lead to gene activation. By comparing the bivalent genes upregulated in either or both mutants, we could identify two subsets: (i) one more susceptible to the decreases in the promoter H3K27me3 levels (comprising lineage marker genes) and affected in both mutants and (ii) a subset less sensitive to H3K27me3 decrease, yet relying for gene expression on the activation of distal regulatory elements and upregulated exclusively in H3.3K9A mESCs, due to the activation of ERV-derived CREs.

Notably, we observed in H3.3K9A mESCs the unusual activation of regulatory networks controlling the expression of immune genes, which also culminates in the expression of proteins typically expressed only in specialized immune cells and in the premature expression of lineage markers in early stages of in vitro macrophage differentiation. Although de-repressed ERVs may produce endogenous viral transcripts, which could trigger an interferon-mediated antiviral response, we did not observe a consistent and robust upregulation of interferon-stimulated genes. This is in line with the reported limited ability of mESCs to mount innate immune responses[54,55]. In addition, treating mESCs with a small-molecule inhibitor of the RIG-I receptor[56], did not result in a consistent downregulation of immune-related genes. Together with the fact that ERVs have been co-opted throughout evolution to function as CREs in immune cells and other differentiated cell types, we envision that the ERV-derived cryptic CREs controlled by H3.3K9me3 in mESCs directly regulate the transcription of specific immune genes and developmental genes subsets, placing histone H3.3 lysine 9 as a critical caretaker of distinct distal genomic regions.

## Methods

### Cell culture
Mouse embryonic stem cells (129XC57BL/6J) were cultured in ESC medium containing Knockout-DMEM (Thermo Fisher Scientific) supplemented with 15% EmbryoMax fetal bovine serum (ES FBS Merck-Millipore) and 20 ng/ml leukemia inhibitory factor (LIF, produced by Protein Expression and Purification Facility at EMBL Heidelberg), 1% non-essential amino acids (Thermo Fisher Scientific), 1% Glutamax (Thermo Fisher Scientific), 1% Pen/Strep (Thermo Fisher Scientific), 1% of 55 mM beta-Mercaptoethanol solution (Merck-Millipore). Upon thawing, cells were cultured on a layer of mouse embryonic fibroblasts (MEFs) and afterward passaged on 0.1% gelatine-coated culture dishes without feeder cells. Cells were maintained at 37 °C with 5% $CO_2$ and routinely tested for mycoplasma. The medium was changed daily, and cells were passed every 2 days.

### CRISPR-Cas9 editing strategy
To introduce mutations of the endogenous *H3f3b* gene, a ribonucleoparticle (RNP) Cas9-based approach was used. First, tracrRNA-ATTO550 and the target-specific crRNA-XT (Integrated DNA Technologies) were resuspended in IDT Nuclease-Free Duplex Buffer to a final concentration of 200 µM. The single-guide RNA (sgRNA; designed using the Benchling Guide RNA Design tool) was assembled in vitro by mixing tracrRNA and crRNA in an equimolar ratio, and the mixture was heated at 95 °C for 5 min and cooled at RT on a bench-top for ~20 min. Second, the Cas9 RNP was assembled by mixing the Cas9-mSA protein (produced by the EMBL Protein Expression and Purification Facility) with the sgRNA in a molar ratio of 1:1.5 Cas9:sgRNA, in a final volume of

10 μl of Neon Buffer R per electroporation reaction and incubated at RT for 10–20 min. The biotinylated repair template (Integrated DNA Technologies Ultramers) was then added to the mix prior to electroporation, considering the same molar ratio of 1:1.5. Cells were washed once with PBS and $10^5$ cells/reaction were resuspended in 10 μl of mix previously prepared (Buffer R + Cas9 RNP complex). The Neon Transfection System (Thermo Fisher Scientific) was used for electroporation using the following program: 1600 V; 10 ms; 3 pulses (from ref. [87]). After electroporation, cells were transferred to a tube containing 500 μl of pre-warmed RPMI medium supplemented with 8% ES FBS and left to recover for 15 min at 37 °C in the incubator. Cells were then centrifuged and washed once with RPMI medium supplemented with 8% ES FBS and plated in a 24-well plate containing MEFs. After 24 h, single-cell sorting was performed to select ATTO550 positive cells and single clones were expanded, genotyped, and validated. The nucleotide sequences of sgRNAs and repair templates used for the editing are reported in Supplementary Data 1.

### Chromosomal integrity check of CRISPR-edited clones
The chromosomal integrity of the homozygous clones generated was assessed via low-coverage whole-genome sequencing. Briefly, genomic DNA was extracted from ~$10^6$ cells using the Gentra Puregene Cell Kit (Qiagen) and sonicated at 4 °C using a Bioruptor Pico sonication device (Diagenode) for ten cycles (30 s ON/30 s OFF) to obtain fragments of ~200 bp. The fragmentation pattern was assessed via agarose gel electrophoresis and 0.5–1 μg of fragmented gDNA was used to prepare sequencing libraries using the NEBNext Ultra II DNA Library Preparation Kit (New England Biolabs), according to the manufacturer's instructions. Libraries were then quantified using the Qubit fluorometer (Thermo Fisher Scientific) and the quality was assessed on a Bioanalyzer using the Agilent High Sensitivity DNA Kit. Samples were then pooled and sequenced on an Illumina NextSeq 500 platform (75 bp single-end reads).

### Low-coverage whole-genome sequencing analysis
The quality of the sequencing run was assessed using FastQC (v0.11.5 – https://github.com/s-andrews/FastQC) and adapter sequences were trimmed using TrimGalore (v0.4.3 – https://github.com/FelixKrueger/TrimGalore). Sequencing reads were aligned to the mouse reference genome (GRCm38/mm10 assembly) using Bowtie2 (v2.3.4[88]) with default settings. Uniquely mapped reads (MAPQ ≥30) were retained for the subsequent steps. The bam files thus generated were then indexed and used for copy number variation analysis (bin size of 100 kb) using the Coral script (https://github.com/tobiasrausch/coral). Preprocessing steps were conducted with the Galaxy platform (v21.09[89]).

### Histones extraction and mass-spectrometry analysis
Histones were extracted, derivatized with propionic anhydride, and digested as previously described[90]. All samples were desalted prior to nanoLC-MS/MS analysis using in-house prepared C18 stage-tips. Histone samples were analyzed by nanoLC-MS/MS with a Dionex-nanoLC coupled to an Orbitrap Fusion mass spectrometer (Thermo Fisher Scientific). The column was packed in-house using reverse-phase 75 μm ID × 17 cm Reprosil-Pur C18-AQ (3 μm; Dr. Maisch GmbH). The HPLC gradient was: 4% solvent B (A = 0.1% formic acid; B = 80% acetonitrile, 0.1% formic acid) for 2 min, then from 4 to 34% solvent B over 48 min, then from 34 to 90% solvent B in 2 min, hold at 90% solvent B for 2 min, and from 90% solvent B down to 4% in 1 min followed by a hold at 4% solvent B for 5 min. The flow rate was at 300 nL/min. Data were acquired using a data-independent acquisition method, consisting of a full scan MS spectrum (m/z 300–1200) performed in the Orbitrap at 120,000 resolution with an AGC target value of 5e5, followed by 16 MS/MS windows of 50 m/z using HCD fragmentation and detection in the orbitrap. HCD collision energy was set to 27, and the AGC target at 1e4. Histone samples were resuspended in buffer A, and 1 ug of total

histones was injected. Histone data was analyzed using a combination of EpiProfile 2.0[91], Skyline[92], and manual analysis with Xcalibur (Thermo Fisher Scientific). The peptide relative ratio was calculated by using the area under the curve (AUC) for that particular peptide over the total AUC of all possible modified forms of that peptide. Data analysis was performed using Microsoft Excel to calculate averages and standard deviations.

### PRC2 in vitro HMT assay
The assay was performed as previously described[93], with the addition of normalization samples that were loaded on each of the gels to allow for a comparison across multiple gels. Each 10 μL HMTase reaction contained 500 nM PRC2 complex, chromatinized DNA with sequence from the ATOH1 locus (see ref. [94]) at a total concentration equivalent to 0.8 μM nucleosomes and 5.0 μM 14C-SAM (S-[methyl-14C]-adenosyl-l-methionine (PerkinElmer, no. NEC363050UC)). All chromatinized DNA constructs were identical except for the H3 histone constructs, which included either H3.1 or H3.3, as indicated, and lysine 9 was either unmodified or modified to include a methyl lysine analog (MLA), as indicated. Each reaction was incubated for 30 min and 2 h at 30 °C in HMTase buffer (50 mM Tris-HCl pH 8.0 (at 30 °C), 100 mM KCl, 2.5 mM MgCl$_2$, 0.1 mM ZnCl$_2$, 2.0 mM 2-mercaptoethanol, and 0.1 mg/mL-1 bovine serum albumin) and then stopped by adding 4× LDS sample buffer (~3.3 μL) (Thermo Fisher Scientific, no. NP0007) to a final concentration of 1× (final volume ~13.3 μL). About 10 μL of each sample was then incubated at 95 °C for 5 min and subjected to SDS-PAGE (16.5% acrylamide). Gels were first stained with InstantBlue Coomassie protein stain (Expedeon, no. ISB1L) and then vacuum-dried for 60 min at 80 °C with the aid of a VE-11 electric aspirator pump (Jeio Tech). Dried gels were exposed to a storage phosphor screen (Cytiva) for 5 days, and the signal was acquired using a Typhoon Trio imager (Cytiva). All experiments were performed in triplicate.

### Growth assay
mESCs were seeded at a density of $10^4$ cells per well in 12-wells format on day 0. Two wells per cell line were trypsinized, harvested and counted each day for a total of 6 days, while the remaining wells received fresh media. Growth curves were generated using the averaged duplicate cell counts.

### Cell cycle assay
The percentage of proliferating cells in the S-Phase was measured using the EdU Click FC-488 Kit (Carl Roth – BA-7779) following the manufacturer's instructions. Briefly, EdU was added at a final concentration of 10 μM, and cells were incubated for 2.5 h at 37 °C. Cells were washed once with PBS and then dissociated using accutase. Cells were fixed by resuspension in 100 μl (every $1 × 10^6$) of 4% PFA in PBS and incubation for 15 min at RT, in the dark. After quenching with PBS + 1% BSA, cells were centrifuged, and the pellet was resuspended in 1X saponin-based permeabilization buffer. A total of $4.5 × 10^5$ cells were used for each condition, and 500 μl of Click reaction master mix (PBS, catalyst solution, dye azide, and 10X buffer additive) were added to each tube. After 30 min of incubation at RT in the dark, cells were washed once with 3 ml of saponin-based permeabilization buffer, and DNA content was stained with propidium iodide (7 min incubation at RT). Cells were washed once with saponin-based permeabilization buffer resuspended in 300–500 μl of buffer and analyzed by flow cytometry. Three controls were included for flow cytometry analysis: (i) w/o EdU, w/o click and w/o DAPI; (ii) w/o Edu, with click and DAPI; (iii) with EdU, w/o click and with DAPI.

### Cell death assay
The FITC Annexin V Kit (BD Pharmingen – 556419) was used, which allows to detect earlier stages of cell death, preceding loss of membrane integrity. Briefly, cells were washed twice with PBS and

resuspended in 1X binding buffer to a concentration of $1 \times 10^6$ cells/ml. To 100 µl of the mix (-$10^5$ cells), 5 µl of FITC Annexin V and 10 µl of propidium iodide (50 µg/ml stock) were added and cells were incubated for 15 min at RT, in the dark. After incubation, 400 µl of 1X binding buffer were added to each tube and cells were analyzed by flow cytometry. Three controls were included for flow cytometry analysis: (i) unstained cells; (ii) cells with FITC Annexin V, w/o PI; (iii) cells w/o FITC Annexin V, with PI.

### Immune proteins staining
Cells were harvested and washed with FACS buffer (PBS + 2% FBS). Cells were then resuspended in 100 µl of master mix composed of FACS buffer and desired antibodies, each one diluted 1:100. Alternatively, cells were resuspended in FACS buffer only for the unstained controls. After 30 min incubation on ice and in the dark, cells were centrifuged and washed with FACS buffer. Antibodies used for the staining were the following: PE-CD59a (143103 – Biolegend); APC-TLR2/CD282 (153005 – Biolegend); FITC-MHC Class II (107605 – Biolegend); PECy7-CD11b (561098 – BD); BV605-F4/80 (123133 – Biolegend).

### Bodipy-C11 staining
The staining was performed following the guidelines described by ref. 95. In brief, the BODIPY-C11 probe was added at a final concentration of 2.5 µM and cells were incubated for 30 min at 37 °C. Cells were then washed with HBSS buffer, collected, and analyzed by flow cytometry.

### Neuronal differentiation
mESCs were differentiated into glutamatergic neurons, as described by ref. 96, with modifications. mESCs were cultured in a differentiation medium containing high-glucose DMEM, 10% FBS (Gibco), 1% non-essential amino acids (Thermo Fisher Scientific), 1% penicillin/streptomycin (Thermo Fisher Scientific), 1% GlutaMAX, 1% sodium pyruvate (Thermo Fisher Scientific), 0.1% of 14.5 M β-mercaptoethanol (Sigma Aldrich) solution. To promote the formation of embryoid bodies, cells were grown in suspension on non-adherent dishes (Greiner Petri dishes – 633102). The differentiation medium was changed every 2 days. From day 4 until day 8, the EBs were treated with 5 µM retinoic acid every 2 days to induce neuronal lineage commitment. On day 8, EBs were dissociated with trypsin and neural progenitor cells (NPCs) were plated on poly-D-lysine hydrobromide (Sigma Aldrich) and laminin (Roche) coated dishes in N2 medium (high-glucose DMEM, 1% N-2, 2% B-27, and 1% penicillin/streptomycin, from Thermo Fisher Scientific) at a density of $2 \times 10^5$ cells/cm². The medium was changed after 2 h and after 24 h. On day 10, the medium was changed to a complete medium (neurobasal, 2% B-27, and 1% penicillin/streptomycin). Glutamatergic neurons were harvested on day 12.

### Mesoderm-cardiomyocyte differentiation
mESCs were differentiated into cardiomyocytes as previously described[18], adapting a protocol from previous publications[97,98]. Briefly, $7.5 \times 10^5$ mESCs were resuspended in differentiation medium (DMEM, 20% FBS, 1% NEAA, 1% P/S, 1% GlutaMAX, and 100 µM ascorbic acid) and grown in suspension on non-adherent dishes to promote EB formation. After 4 days, EB were plated onto 0.1% gelatine-coated plates. The medium was changed every 2 days. Formation of contracting colonies was observed after 8 days of differentiation and time-course quantification of contracting colonies was performed at days 8, 10, and 12. Cardiomyocytes were harvested by trypsinization on day 14.

### Quantitative real-time PCR
RNA was extracted from -$10^6$ cells using RNeasy Kit (Qiagen), followed by DNase digestion using TURBO DNase (Thermo Fisher Scientific). A total of 1 µg of RNA was reverse transcribed with random primers to

cDNA using a High-Capacity cDNA Reverse Transcription Kit (Applied Biosystems). For qRT-PCR reaction, 6 ng of cDNA were used as a template and reactions were performed using Power SYBR Green master mix (Applied Biosystems) on a StepOnePlus Real-Time PCR machine. The comparative Ct method (ΔΔCt method) was used to calculate normalized gene expression values. Ct values of target genes were normalized to Ct values of the housekeeping gene Rpl13 to obtain ΔCt values and to control samples to obtain ΔΔCt values. Primers used for RT-qPCR are listed in Supplementary Data 2.

### Immunofluorescence
NPCs were seeded on glass coverslips at day 8 of neuronal differentiation and fixed at day 12 for neuronal staining. Cells were washed with PBS briefly and fixed with 3% paraformaldehyde (PFA) (Electron Microscopy Sciences) in PBS for 20 min at RT, then quenched with 30 mM glycine in PBS for 5 min at RT. Cells were washed three times with PBS and stored in PBS at 4 °C until needed. Permeabilization was conducted with 0.1% Triton-X in PBS for 5 min at RT, followed by blocking with 0.5% BSA in PBS for 30 min at RT. Primary antibody incubation was performed for 1 h at RT under constant shaking. The following antibodies were diluted 1:200 in 0.5% BSA and used for stainings: Map2 (Sigma Aldrich, 9942) and β-III tubulin (Abcam, ab78078). Cells were washed three times with PBS and incubated for 30 min at RT with a secondary antibody (goat anti-mouse IgG Alexa 594, Thermo Fisher Scientific – A11005), diluted 1:1000 in 0.5% BSA. After washing twice with PBS, cells were counterstained with 5 µg/ml DAPI for 5 min at RT and washed three times with PBS. Coverslips were mounted with Mowiol mounting medium (Calbiochem) and imaged on a Nikon Eclipse Ti fluorescence microscope. Images were processed with Fiji ImageJ.

### Macrophage differentiation
mESCs were differentiated into macrophages, as described by ref. 36, with modifications. Briefly, $5 \times 10^5$ cells were seeded in 6 cm non-adherent Greiner dish, with 4 ml of differentiation medium (DMEM high glucose, 10% FBS, 15% L929 conditioned medium, 1% NEAAs, 1% GlutaMAX, 1% Na-Pyruvate, 1% β-Mercaptoethanol, and 1 ng/ml of IL-3). On day 4 and day 6, embryoid bodies (EBs) were transferred to new dishes with fresh differentiation medium. On day 8, EBs were transferred to gelatin-coated dishes to favor their attachment and the formation of Factories. Every 2 days, and up until day 20, free-floating macrophage precursors produced from the Factories were transferred into new dishes for their further differentiation with macrophage medium (DMEM high glucose, 10% FBS, 15% L929 conditioned medium, 1% NEAAs, 1% GlutaMAX, 1% Na-Pyruvate, 1% β-Mercaptoethanol, and 1% Pen/Strep). Four days post-harvesting, mature ES cells-derived macrophages (ESDMs) were collected for their purpose.

### RIG inhibitor treatment
The RIG012 small-molecule inhibitor published by ref. 56 was used for this experiment. In particular, control and mutant mESCs were treated for 5 h with RIG012 at a final concentration of 2.5 µM (dissolved in 0.8% DMSO) or with vehicle control. Cells were then collected, RNA was extracted using RNeasy Kit (Qiagen) in combination with RNase-Free DNase Set (Qiagen), and cDNA was generated with random primers using High-Capacity cDNA Reverse Transcription Kit (Applied Biosystems). RT-qPCR was performed as described above (see "Quantitative real-time PCR" section).

### PiggyBac rescue experiments
Stable expression of histone H3 constructs in K9A mESCs was achieved using the PiggyBac Transposon system. The coding sequences of wild-type H3.3, H3.3K9A, and H3.1 were cloned into the donor PB-EF1α-MCS-IRES-RFP vector (System Biosciences, #PB531A-1). A total of $2 \times 10^6$ mESCs were co-transfected with Super PiggyBac Transposase and donor vectors in a 1:2.5 ratio, using the Lonza 4D-Nucleofector X

Unit with 100 μl cuvettes. A bulk of RFP-positive cells were selected by flow cytometry on day 2 after transfection, and again on day 10 to select cells that had stably integrated histone H3 constructs. Each biological replicate of K9A mESCs was transfected with the three donors independently. For H3K9me3 ChIP-qPCR and RT-qPCR experiments, the three independent replicates for control, K9A, K9A + H3.3K9A, K9A + H3.3, and K9A + H3.1 mESC lines were used as detailed in "Cross-link ChIP", "ChIP-qPCR", and "Quantitative real-time PCR" sections. For immunostaining and FACS analysis, two replicates of K9A rescue lines were employed, using the protocol previously described (see "Immune proteins staining" section).

## mRNA-seq

RNA was extracted from ~$10^6$ cells using the RNeasy Mini kit (Qiagen) and DNase digestion was performed using TURBO DNase (Thermo Fisher Scientific), according to the manufacturer's instructions and an additional clean-up of the RNA was then performed using the RNeasy Mini kit (Qiagen). The quality of the RNA was assessed on Bioanalyzer using the Agilent RNA 6000 Nano Kit and samples with an RNA integrity number (RIN) of 10 were used for library generation. For each sample, 100 ng of total RNA was used as input for mRNA selection and conversion to cDNA using the NEBNext Poly(A) Magnetic Isolation Module (New England BioLabs). Sequencing libraries were prepared using the NEBNext Ultra II RNA Library Preparation Kit for Illumina (New England BioLabs), according to the manufacturer's instructions. After Qubit quantification, the quality of individual libraries was assessed on a Bioanalyzer using the Agilent High Sensitivity DNA Kit. Samples were then pooled and sequenced on an Illumina NextSeq 500 sequencer (read length of 75 bp in single-end mode).

## mRNA-seq analysis

The quality of the sequencing reads was assessed using FastQC (v0.11.5 – https://github.com/s-andrews/FastQC) and adapter sequences were trimmed using TrimGalore (v0.4.3 – https://github.com/FelixKrueger/TrimGalore). Sequencing reads were aligned to the mouse reference genome (GRCm38/mm10 assembly) using STAR (v2.5.2b[99]) with default settings. Uniquely mapped reads (MAPQ ≥20) were retained for the subsequent steps. Gene count tables were generated with featureCounts (subread v1.6.2[100]), using gencode gene annotations (release M10). Coverage files were generated using bamCoverage (deeptools v2.4.1[101]). Differential expression analysis was performed in R using the DESeq2 package[102]. Genes were considered differentially expressed using a false discovery rate (FDR) cut-off of 0.05, and the fold-change cut-off applied is indicated in the main text and in figure captions. MA plots were generated using the ggmaplot function (from the ggpubr R package – v0.4.0; https://github.com/kassambara/ggpubr). Upset plot in Fig. 1e was generated using the UpSetR package[103]. Gene ontology enrichment analysis was conducted using ClusterProfiler[104]. Other plots were generated using the ggplot2 R package (https://github.com/tidyverse/ggplot2). Pre-processing steps were conducted with the Galaxy platform (v21.09).

## Transposable elements expression analysis

For transposable elements (TEs) expression analysis, the RepEnrich tool[48] was used in its updated version (https://github.com/nerettilab/RepEnrich2). Sequencing reads were aligned to the mouse reference genome (GRCm38/mm10 assembly) using Bowtie2 (v2.3.4), and retaining secondary alignments. LTRs and non-LTRs annotation for the mm10 mouse genome assembly were retrieved using the RepeatMasker track from the UCSC genome table browser and prepared as suggested. After running RepEnrich2 independently for all samples, the individual output tables with estimated counts were merged, and the differential expression analysis for LTRs and non-LTRs was performed in R using DESeq2. Repeats were considered significantly differentially expressed using a FDR cut-off = 0.05, and an abs(log2 fold-change) cut-

off = 1.5. The PCA plot was generated in R with ggplot2 and volcano plots were generated with the EnhancedVolcano package (https://github.com/kevinblighe/EnhancedVolcano). For the overlap analyses (Fig. 5a) and for inspection of specific loci on the genome browser (Figs. 6a, b, 7i, and Supplementary Figs. 9, 11e), only ERVs with size ≥500 bp were considered.

## Native ChIP-seq

DNA for native ChIP was digested by MNase treatment to obtain mainly mono-nucleosomes using a modified protocol from ref. 105. For each IP, $20 \times 10^6$ cells were resuspended in digestion buffer (50 mM Tris-HCl pH 7.6; 1 mM $CaCl_2$; 0.2% Triton-X), treated with 100 U MNase (Worthington) and incubated at 37 °C for 5 min while shaking at 500 rpm. Samples were quickly moved to ice and MNase was quenched by the addition of 5 mM EDTA (final). Lysates were sonicated for three cycles using Bioruptor Pico (Diagenode) and dialyzed against RIPA buffer for 3 h at 4 °C. Insoluble material was pelleted at maximum speed for 10 min at 4 °C and supernatant was used as input for ChIP. To check the fragmentation pattern, input DNA was analyzed by agarose gel electrophoresis. A 5% fraction of input was set aside for sequencing. Protein G-Dynabeads (Invitrogen) were pre-coated with antibodies for 4 h at 4 °C and the following antibodies were used: H3K27ac (39685 – Active Motif); H3K9ac (C5B11-9549 – Cell Signalling Technology). After overnight incubation of the pre-coated beads with chromatin lysates, beads were washed (3X with RIPA; 2X with RIPA + 300 mM NaCl; 2X with LiCl buffer–250 mM LiCl, 0.5% NP-40, 0.5% NaDoc) and finally rinsed with TE + 50 mM NaCl Buffer. Samples were eluted from Protein G Dynabeads using SDS elution buffer (50 mM Tris-HCl ph 8.0; 10 mM EDTA; 1% SDS) at 65 °C for 30 min with shaking at 1500 rpm. Proteinase K was added (0.2 μg/ml final) to the eluted samples and digestion was carried out for 2 h at 55 °C followed by PCR purification (Qiagen). Sequencing libraries were prepared using the NEBNext Ultra II DNA Library Preparation Kit (New England Biolabs) and sequenced on the Illumina NextSeq 500 platform (75 bp in single-end mode).

## Cross-link ChIP-seq

The protocol was adapted from ref. 39 and ref. 38. Cells were harvested and cross-linked in 3 ml of pre-tempered (25 °C) ES medium supplemented with 1% formaldehyde ($10^6$ cells/3 ml) for 10 min at RT, with rotation. Formaldehyde was then quenched with glycine (125 mM final) and incubated for 5 min at RT, with rotation. Cells were washed twice with ice-cold PBS containing 10%FBS (centrifugation at 200 × g for 4 min, at 4 °C). Pellets can be stored at −80 °C for several months. To prepare lysates, fixed cells were resuspended in 300 μl of Sonication Buffer 1 (50 mM Tris-HCl pH 8.0; 0.5% SDS) and incubated for 10 min on ice. Sonication was performed with Bioruptor Pico (Diagenode) for 20 cycles (30″ON/30″OFF). Sonicated lysates were then diluted 1:6 with lysis buffer (10 mM Tris-HCl ph 8.0; 100 mM NaCl; 1% Triton-X; 1 mM EDTA; 0.5 mM EGTA; 0.1% NaDoc; and 0.5% N-laurolsarcosine) and the soluble fraction was collected after centrifugation at full speed for 10 min at 4 °C. A 2.5% fraction of the supernatant was set aside for input, and the fragmentation pattern (~150–500 bp) was checked by agarose gel electrophoresis. Lysates can be aliquoted and stored at −80 °C with 10% glycerol (final). For each IP, 30 μl Protein G Dynabeads (Invitrogen) were washed twice with 1 ml PBS-T (PBS + 0.01% Tween-20) and incubated with the desired antibody for at least 1 h at RT (or >1 h at 4 °C). Antibodies used with this protocol were the following: H3.3 (09-838 – Merck-Millipore); H3K9me3 (D4W1U – Cell Signalling Technology); H3K9me2 (D85B4 – Cell Signalling Technology); H3K27me3 (C36B11 – Cell Signalling Technology); H3K18ac (9675 – Cell Signalling Technology); SUZ12 (D39F6-3737 – Cell Signalling Technology).

Coated beads were then washed (1X with PBS-T; 2X with lysis buffer), resuspended in 30 μl of lysis buffer/IP and added to the desired amount of chromatin (generally 15–25 μg of chromatin were used, corresponding to 2–4 × $10^6$ cells). After overnight incubation at 4 °C

with rotation, beads-immunocomplexes were washed twice—each time for 5 min—with the following buffers: RIPA; RIPA + 360 mM NaCl; LiCl buffer (10 mM Tris-HCl ph 8.0; 250 mM LiCl, 0.5% NP-40, 0.5% NaDoc; 1 mM EDTA) and finally quickly rinsed with TE buffer and eluted in ChIP SDS elution buffer (10 mM Tris-HCl ph 8.0; 300 mM NaCl; 5 mM EDTA; 0.5% SDS). RNA/protein digestion was performed on beads by adding 2 µl RNaseA (10 mg/ml stock) incubated 30 min at 37 °C, followed by addition of 1.5 µl Proteinase K (20 mg/ml stock) incubated 1 h @ 55 °C. Reverse cross-link was performed with overnight incubation at 65 °C. DNA was purified with 1.4X SPRI-select beads. For H3K27me3 ChIP-Rx, spike-in chromatin was prepared from Drosophila Schneider 2 (S2) cells following the same protocol, and aliquots were stored at −80 °C with 10% glycerol. Before immunoprecipitation, 3% of exogenous chromatin was added to each reaction. Sequencing libraries were prepared using the NEBNext Ultra II DNA Library Preparation Kit (New England Biolabs) and sequenced on Illumina NextSeq 500 (75 bp in single-end mode) or NextSeq2000 (P3 kit – 88 bp in single-end mode).

### ChIP-seq analysis

The quality of the sequencing run was assessed using FastQC (v0.11.5 – https://github.com/s-andrews/FastQC) and adapter sequences were trimmed using TrimGalore (v0.4.3 - https://github.com/FelixKrueger/TrimGalore). Sequencing reads were aligned to the mouse reference genome (GRCm38/mm10 assembly) using Bowtie2 (v2.3.4). Uniquely mapped reads (MAPQ ≥20) aligning to major chromosomes were retained for the subsequent steps. For H3K9me3 ChIP-seq, ~50% of the reads did not map uniquely. ChIP signal strength and sequencing depth were assessed using the plotFingerprint and plotCoverage (deeptools v2.4.1). Coverage files (bigwig format) were generated using deeptools bamCoverage: 10 bp was used as bin size, the "reads per genomic content (RPGC)" method was used for normalization, and reads were extended to an average fragment size of 150 bp. For the H3K27me3 ChIP-seq with exogenous spike-in, reads were aligned to a mouse (mm10) + Drosophila (dm6) combined reference genome. The number of uniquely mapped reads aligning to major mouse and fly chromosomes was retrieved and used to calculate normalization factors that were used with deeptools bamCoverage to generate scaled bigwig files. Peak calling was performed with MACS2[106], by providing ChIP and respective input bam files and considering a minimum FDR cut-off for peak detection of 0.05; narrow or broad peak calling was performed for histone modifications following ENCODE guidelines (https://www.encodeproject.org/chip-seq/histone/). Differential peak analysis was performed in R using the DiffBind package[107], only peaks detected in at least two replicates were retained for the analysis, and peaks were called as significantly differential considering an FDR cut-off = 0.05. Annotation of ChIP-seq peaks was performed with the ChIPseeker package[108] and with publicly available ChromHMM state maps for mESCs (https://github.com/guifengwei/ChromHMM_mESC_mm10).

Annotations for bivalent promoters were obtained using previously generated H3K4me3 and H3K27me3 ChIP-seq data[18]. Briefly, a consensus peakset was defined for the control mESCs (peaks identified in at least two replicates) for both datasets; regions of overlap for the two histone marks were obtained using findOverlaps-methods and genomic regions within a 500 bp window were merged. Genomic coordinates of promoters of protein-coding genes were retrieved from ENSEMBL using biomart (mm10 version: https://nov2020.archive.ensembl.org), and only H3K4me3-H3K27me3 regions overlapping with TSS ± 1 kb were retained. This annotation was further refined by selecting promoters overlapping with peaks of PRC2 subunits (i.e., SUZ12; EZH2; EZH1; EED) retrieved from the ChIP-Atlas database[24] and finally validated with ChromHMM state maps for mESCs. Custom bivalent genes annotation is included in the Source Data file. Heatmaps and metagene plots were generated in R: genomic regions of interest

were binned in 40 windows of 250 bp each, the ChIP-seq signal from individual bigwig files was measured in each window, and all bigwig regions overlapping the same window were averaged. The signal for replicates in each condition was then averaged. Heatmaps were generated using the pheatmap package (https://cran.r-project.org/web/packages/pheatmap/index.html). Boxplots and violin plots were generated considering the ChIP-seq signal measured at TSS or peak summit ± 1 kb. Overlaps between genomic regions of interest were quantified using the GenomicRanges objects and findOverlaps-methods.

Genome browser shots in Figs. 6a, b, 7i and Supplementary Figs. 9, 11e were generated using the Gviz package[109]; bigwig files used for the visualization represent the average ChIP-seq signal from the biological replicates of each genotype, which was computed using WiggleTools[110].

### ChIP-Atlas data integration

Publicly available ChIP-seq data for chromatin factors and transcription factors in mESCs (mm10 assembly) were retrieved from the ChIP-Atlas database[24] and the analysis was performed with an approach similar to the one previously described in ref. 111. Briefly, genomic coordinates of peaks for all mESCs factors (n = 292) were retrieved from the ChIP-Atlas Peak Browser, and the findOverlaps function was used to overlap these peaks with genomic regions of interest. The significance of the relative enrichment of chromatin/transcription factors was assessed using Fisher's exact test. Factors were selected as follows: (i) significant in DA regions versus non-DA regions (randomly selected H3K9me3 and H3K27ac genomic regions considered as background), (ii) significant in each cluster versus other clusters; only factors significant in both comparisons were retained and used for STRING network analysis.

### Transposable elements ChIP-seq analysis

Analysis of H3.3, H3K9me3, and H3K27ac ChIP-seq data to detect enrichment at transposable elements was carried out using the Transposable Element Enrichment Estimator (T3E) tool[51]. Sequencing reads were aligned to the mouse reference genome (GRCm38/mm10 assembly) using Bowtie2 (v2.3.4), retaining secondary alignments. T3E was run for all the ChIP replicates and input samples, filtering out genomic regions of high signals (as suggested by the developers for the mouse genome). A total of 20 iterations were performed when running T3E and calculating enrichments. ChIP-seq enrichment at TEs was considered significant with a $p$ value <0.01 and ERVs with high H3.3 occupancy were additionally filtered by applying a log2(fold-change) >1 cut-off.

### ChIP-qPCR

Prepared libraries from ChIP experiments were diluted and 5 ng of DNA was used as input for each qPCR reaction with SYBR Green PCR Master Mix (Applied Biosystems). The qPCR reactions were performed on a StepOnePlus Real-Time PCR machine. For each condition, biological replicates (ChIP material from two independent mutant cell lines) were used and measured in technical duplicates. Primers used for H3K9me3 and H3K18ac ChIP-qPCR are listed in Supplementary Data 2.

### PRO-seq

Precision nuclear run-on transcription sequencing (PRO-seq) experiments were performed following the protocol previously described[112]. PRO-seq was performed in two batches, one with two replicates each of control and H3.3K27A mESCs and one with three replicates each of control and H3.3K9A mESCs. Briefly, $10^7$ mESCs were permeabilized and used for each reaction, and 5% (i.e., 500k) of permeabilized Drosophila S2 cells were added prior to the in vitro run-on reaction. Run-on reactions were carried out by adding 100 µl of 2× NRO reaction mixture (10 mM of Tris-HCl, pH 8.0; 5 mM MgCl$_2$; 300 mM KCl; 1 mM

dithiothreitol (DTT); 1% sarkosyl; 375 μM biotin-11-CTP/-UTP; 375 μM ATP/GTP; 2 μl RNase inhibitor) to 100 μl of the permeabilized cells, and incubated at 30 °C for 3 min. The addition of TRIzol LS (Thermo Fisher Scientific) terminated the reaction and RNA was extracted and precipitated in 75% ethanol. Nuclear RNA was fragmented by base hydrolysis in 0.2 N of NaOH on ice for 10 min and subsequently neutralized by adding the same volume of 1 M of Tris-HCl, pH 6.8. Fragmented biotinylated RNA was bound to the streptavidin magnetic beads (Thermo Fisher Scientific), washed, eluted using TRIzol, and precipitated in ethanol. Following 3′–RNA and 5′–RNA adapter ligation, the precipitated RNA was reverse-transcribed and PCR-amplified. DNA libraries were purified using SPRI-select beads and quality was assessed on a Bioanalyzer using the Agilent High Sensitivity DNA Kit, pooled and sequenced on an Illumina NextSeq 500 platform (75 bp in single-end mode).

### PRO-seq analysis

The analysis of the PRO-seq data was performed according to the guidelines provided by the Danko Laboratory. In particular, pre-processing and alignment was performed using the PROseq2.0 pipeline (https://github.com/Danko-Lab/proseq2.0). The consensus set of distal *cis*-regulatory elements active at the basal state in mESCs was obtained by running the dREG package (https://github.com/Danko-Lab/dREG) on the PRO-seq data from control cell lines. The differential PRO-seq analysis was performed using the output of the dREG package from the control and H3.3K9A lines as input for the tfTarget package (https://github.com/Danko-Lab/tfTarget). The violin plots in Figs. 3e, 6e and S10f were generated by calculating RPKM values for mutant and control samples within the genomic regions of interest.

### Homer TF motifs analysis

For transcription factor motifs analysis, we employed the Homer tool with different settings, according to the input sequences used. Specifically, for TF motif enrichment analysis at Cluster 1 H3K9me3 regions and at canonical enhancers (Fig. 6c, d), we selected active regions identified through the dREG package and performed de novo motif enrichment analysis using Homer with the "–size 200" parameter, to focus on the center of each PRO-seq peak. For TF motif enrichment analysis at ERV sequences (Supplementary Fig. 7k, l), we retrieved from RepeatMasker the genomic coordinates of each repeat instance belonging to the ERV subfamilies of interest. The generated BED files were used as input for Homer de novo motif analysis with the "–size given" parameter, to take into account the entire ERV sequence.

### Gene regulatory network construction and differential transcription factor activity analysis

The GRN was generated as described in ref. 25, integrating mRNA-seq and H3K27ac ChIP-seq datasets obtained from 12 independent mESCs samples (i.e., three biological replicates each for Control, H3.3K9A, H3.3K27A, and H3.3K79A mESCs, respectively) which were used to construct the GRN. An FDR cut-off = 0.2 was used for peak-gene connections, yielding a GRN with TF-CRE-gene connections composed of a network of 153 unique TFs, 5694 regulatory elements, and 5229 genes (of which 1194 DE) overall resulting in 16,701 CRE-gene connections.

The differential transcription factor activity analysis was performed using the diffTF method described in ref. 29, using the same 12 mRNA-seq and ChIP-seq datasets as input.

### Reporting summary

Further information on research design is available in the Nature Portfolio Reporting Summary linked to this article.

## Data availability

The DNA-seq, mRNA-seq, ChIP-seq, and PRO-seq data generated in this study have been deposited in the ArrayExpress database under the following accession codes: E-MTAB-12866 (mRNA-seq), E-MTAB-12867 (DNA-seq), E-MTAB-12868 (ChIP-seq), and E-MTAB-12869 (PRO-seq). The histone modifications mass-spectrometry raw data have been deposited to the Proteome Xchange repository via the MassIVE database and are available through the identifier PXD053371 [http://proteomecentral.proteomexchange.org/cgi/GetDataset?ID=PXD053371]. Source data are provided with this paper.

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

## Acknowledgements

We thank the staff of the European Molecular Biology Laboratory (EMBL) Genomics Core Facility, Protein Expression and Purification Facility, and Flow Cytometry Core Facility for sample preparations and data generation. We thank Charles Girardot and the Genome Biology Computational Support for their assistance in data analysis and submission. We thank Mohamad Whaidi, Maja Gehre, and Nichole Diaz for helping with mutant line generation. We thank Andrea Callegari and Moritz Kueblbeck for sharing their expertise in CRISPR-Cas9 gene editing. We thank Michael Bonadonna and Diana Ordonez for helping with flow cytometry stainings. We thank Brian Mang Ching Lai for helping with the GRN construction. We thank Elena Vizcaya Molina for providing the Drosophila S2 cells used as spike-in in PRO-seq and ChIP-seq experiments. We thank Guy Riddihough for critical feedback on this manuscript and Eileen Furlong and the Noh laboratory for the helpful discussions. This work was supported by the EMBL predoctoral fund (to K.-M.N), the DFG fund (SPP 1738 to K.-M.N.), the EIPOD postdoctoral fund (to D.B.), and the EMBL collaborative grant (MeH3 to K.M.N and C.D.). B.A.G. gratefully acknowledges the NIH grants.

## Author contributions

M.T. and K.-M.N. conceived the project. M.T. and K.-M.N. designed the experiments. M.T., D.B., U.Y., N.F.-N.M., and A.J. collected and analyzed the data. M.T. and D.B. performed the bioinformatic analyses. J.B.Z. supervised D.B. M.U., V.L., and C.D. designed and performed the PRC2 in vitro assays. Y.P. performed the histone modifications mass-spectrometry analysis with supervision from B.A.G. M.T. generated the figures. M.T. and K.-M.N. wrote the manuscript with input from all authors.

## Funding

## Competing interests

The authors declare no competing interests.
