## [Peer Review File · Nature Communications]

Histone H3.3 lysine 9 and 27 control repressive chromatin at cryptic enhancers and bivalent promotersREVIEWER COMMENTS

Reviewer #1 (Remarks to the Author):

The authors generated and characterized H3.3K9A and H3.3K27A mutant mouse ES cells to examine their effects on gene regulation and chromatin organization. It turns out that H3.3K9A mutant cells have significant defects in transcriptional regulation and dysregulate a large number of genes (both upregulated and downregulated). These cells also exhibit defects in differentiation and are characterized by growth defects, linked to higher apoptosis.

Chromatin organization is altered by impaired H3K9me3 occupancy on H3.3-H3K9me3 target regions, whereas H3.1/2-H3K9me3 regions do not seem to be affected. These data are consistent with the role of H3.3 in heterochromatin organization and highlight the importance of the H3K9 position on H3.3 in this context.

Furthermore, the authors correlate changes in gene expression (particularly upregulated genes) with the loss or reduction of H3K9me3 in the vicinity of these target genes. These H3K9me3 regions contain putative cis regulatory elements and sequences related to endogenous retroviruses that may abnormally act as enhancers in the H3.3K9A mutant cells. Finally, the authors observed that H3K27me3 is affected by both the H3.3K27A and H3.3K9A mutants on bivalent genes, suggesting that H3.3K9 plays a role in PCR2 activity.

The finding that H3.3K9A affects H3K9 establishment on H3.3-H3K9me3 target regions is interesting; however, the authors do not provide any novel mechanistic insight into this process. The connection between dysregulated H3K9me3 (on ERVs and inactive CREs) and changes in gene expression has been shown in many publications and does not reveal novel insights in the present study. Therefore, the manuscript is largely descriptive overall.

The following points may improve the study:

1) The role of H3.3 in heterochromatin organization has been established in several studies. The authors should contextualize their findings within these previous studies and propose mechanistic models to explain the effects on H3K9me3 establishment in their system.

2) The extent of transcriptional dysregulation in H3.3K9A seems quite large compared to Daxx/Atrx/H3.3 ko cells, which impair H3.3 deposition on heterochromatic regions. This raises the question of direct vs. indirect effects. Transcriptional changes may occur due to aberrant H3.3K9A on physiological regulatory regions, impaired heterochromatin, and changes in the transcription factor network resulting from these effects. Establishing cause-consequence relationships is challenging but would provide deeper mechanistic insights. In particular, conducting rescue experiments to directly relate transcriptional changes with the H3.3K9A mutation would be important.

Additional points:

* It is not clear if independent mutant clones were analyzed, if they exhibit the same phenotype,

and if the experiments were replicated using independent clones.

* The deposited NGS data could not be accessed, making it impossible to assess the data quality.

* It would be beneficial to include more IGV screenshots to gain a better impression of the data, including H3.3 tracks.

* Figure S4d shows gene dysregulation in the vicinity of H3K9me3 target loci. It appears that there is significant regulation in the non-changed H3K9me3 regions. This finding could actually support the notion that indirect effects are major contributors to the observed transcriptional changes.

* H3.3K27A knockout cells were previously published by the Noh lab. Are these the same cells?

Reviewer #3 (Remarks to the Author):

To investigate the role of H3.3 K9 and K27 PTM, authors mutated these two residues in mouse mESCs and characterized the epigenomic landscape of these cells.

Taking advantage of an initial RNA-seq experiment, they identified GO categories of differentially expressed genes. Importantly, upregulated genes in K9A were related mainly to immune response while upregulated genes in K27A and in both K9A and K27A mutants were commonly enriched for development and differentiation processes.

Noting that some of the genes were known markers of embryonic lineages, they applied in vitro differentiation protocols to find that K9A mESCs failed to differentiate in neurons, induced cardiac-mesoderm EBs failed to progress further differentiation and cells showed increased apoptosis.

They then carried out ChIP-seq experiments identifying differential peaks that could be clustered in two groups:

Cluster 1 lost the H3K9me3 signal in the K9A mutant, presented a higher basal level of H3K27ac in control mESCs, which was further increased in K9A mESCs. It showed a marked gain in nascent transcription and a relative enrichment of transcriptional co-activators, subunits of chromatin remodelers and TFs linked to pluripotency and development.

Cluster 2 had no basal histone H3 acetylation, a modest increase in H3K27ac and H3K9ac in K9A mESCs and a relatively higher enrichment for core heterochromatin factors, such as HP1 proteins and other transcriptional repressors.

They then showed that a significant fraction of the cryptic CREs de-repressed in H3.3K9A mESCs, function as enhancers active in specialized immune cell types. Importantly, a large fraction of the cryptic CREs activated in H3.3K9A mESCs is ERV-derived enhancers, and some of them are capable of driving the expression of immune-response genes.

Finally, they propose a model where in K9A mESCs, de-repressed ERV-derived CREs upregulate a subset of genes and potentially provide a plethora of TFs binding sites resulting in a redistribution of TFs and subsequent gene expression changes.

The paper is interesting and addresses a very important biological problem in the field. Experiments and data analysis are well done and results are compelling.

There are however some crucial points that remain unclear and need to be addressed.

First of all, the paper ignores what is already known about the role of ERV in ESCs. There are several

papers that have studied this important issue (see i.e. ; doi:10.1038/nsmb.2799; doi:10.1038/ng.600; doi:10.1038/ng.2965).

Please analyze and discuss these data under the light of previous knowledge.

What is happening to the expression of ERV elements that are believed to be expressed and functional in ESC? What is the interplay, commonalities and differences between ERV involved in ESC maintenance and differentiation in immune cell types? Any difference in TFBS?

In general what is missing is a general characterization of Transposable elements expression in any of the data analysis that has been carried out in the paper.

What is the basic level of expression of TE in general and ERV in particular in WT cells and what is their expression in edited cells? Please identifies TE families that are differentially regulated by using ad hoc tools i.e. Square.

How much of your H3K9me3 ChIP-seq signal is unmappable?

Given the experience in differentiating cell lineages in authors' laboratory and the knowledge accumulated in this study, an experiment of in vitro differentiation of functional immune cell types should be required as a final validation of the functional implications of reactivation of cryptic enhancers for the immune system.

A more detailed description of TFs linked to pluripotency in Cluster 1 should be presented.

We sincerely appreciate the reviewers for their constructive feedback, which has greatly enhanced our manuscript. Below, we provide our point-by-point responses alongside figures containing panels derived from the new analyses and experiments performed. The changes incorporated into the revised manuscript are marked in *Italic* font within each point-by-point response and as purple-colored text within the revised manuscript.

REVIEWER COMMENTS

Reviewer #1 (Remarks to the Author):

The authors generated and characterized H3.3K9A and H3.3K27A mutant mouse ES cells to examine their effects on gene regulation and chromatin organization. It turns out that H3.3K9A mutant cells have significant defects in transcriptional regulation and dysregulate a large number of genes (both upregulated and downregulated). These cells also exhibit defects in differentiation and are characterized by growth defects, linked to higher apoptosis.

Chromatin organization is altered by impaired H3K9me3 occupancy on H3.3-H3K9me3 target regions, whereas H3.1/2-H3K9me3 regions do not seem to be affected. These data are consistent with the role of H3.3 in heterochromatin organization and highlight the importance of the H3K9 position on H3.3 in this context.

Furthermore, the authors correlate changes in gene expression (particularly upregulated genes) with the loss or reduction of H3K9me3 in the vicinity of these target genes. These H3K9me3 regions contain putative cis regulatory elements and sequences related to endogenous retroviruses that may abnormally act as enhancers in the H3.3K9A mutant cells. Finally, the authors observed that H3K27me3 is affected by both the H3.3K27A and H3.3K9A mutants on bivalent genes, suggesting that H3.3K9 plays a role in PRC2 activity.

The finding that H3.3K9A affects H3K9 establishment on H3.3-H3K9me3 target regions is interesting; however, the authors do not provide any novel mechanistic insight into this process. The connection between dysregulated H3K9me3 (on ERVs and inactive CREs) and changes in gene expression has been shown in many publications and does not reveal novel insights in the present study. Therefore, the manuscript is largely descriptive overall.

Response: We appreciate the reviewer's interest in our findings. It is worth noting that we have elucidated the controversial role of H3.3K9 in heterochromatin organization, function, and cellular physiology (further details are provided below). Our contribution lies in providing mechanistic insights into this process by identifying H3.3K9me3-marked cis-regulatory regions and their functional significance. In response to the reviewer's suggestions, we conducted rescue experiments, yielding more robust evidence that these regions directly regulate developmental genes and immune-associated genes. Furthermore, our investigations have uncovered that H3.3, particularly when the H3.3K9 residue is unmodified, serves as a superior substrate for PRC2 activity compared to H3.1, thereby shedding light on the mechanistic role of the H3.3K9 residue.

The following points may improve the study:

1) The role of H3.3 in heterochromatin organization has been established in several studies. The authors should contextualize their findings within these previous studies and propose mechanistic models to explain the effects on H3K9me3 establishment in their system.

Response: The reviewer is right. To address this point and the subsequent one (the extent of transcriptional dysregulation in H3.3K9A seems quite large compared to Daxx/Atrx/H3.3 ko cells), we have included the following paragraph in the “Discussion” section (page 20). We are grateful to the reviewer for raising this concern.

“H3.3 has been associated with heterochromatin organization due to increased activation of heterochromatin elements such as ERVs, observed in H3.3 knockout mESCs or H3.3 chaperon knockout mESCs [1–3]. However, the precise contribution of H3.3, specifically H3.3K9, to heterochromatin function remains unclear [2–4]. By introducing the K9A substitution in the endogenous H3.3, we investigated its impact on chromatin landscape and transcription. We found that the removal of the H3.3K9 residue led to a decrease in the H3K9me3 signal at specific genomic regions (about 18% of total H3K9me3 domains), particularly at ERVK (IAPs, IAPLTRs, IAPes) and ERV1 (RLTR1s), aligning with the result in H3.3 knockout mESCs [1]. Rescue experiments confirmed the central role of H3.3K9 in H3K9me3 accumulation and transcriptional repression, particularly at IAPs.

Furthermore, we demonstrated the essential role of H3.3K9 in suppressing gene regulatory activities associated with specific ERVs, leading to abnormal activation of transcriptional programs in K9A mESCs. Compared to H3.3 knockout mESCs, H3.3K9A mESCs displayed more prominent transcriptional dysregulation, likely due to the preservation of histone variant incorporation into chromatin. Additionally, unlike KAP1/TRIM28 knockouts, which induce significant ERV activation [5, 6] but exhibit a rapid decline in mESCs viability [7, 8], H3.3K9A mESCs can be maintained in culture, facilitating the comprehensive investigation of the transcriptional rewiring upon ERV activation. Thus, the H3.3K9A approach offers unique insights distinct from H3.3 knockout or H3.3 chaperon knockout methods.”

2) The extent of transcriptional dysregulation in H3.3K9A seems quite large compared to Daxx/Atrx/H3.3 ko cells, which impair H3.3 deposition on heterochromatic regions. This raises the question of direct vs. indirect effects. Transcriptional changes may occur due to aberrant H3.3K9A on physiological regulatory regions, impaired heterochromatin, and changes in the transcription factor network resulting from these effects. Establishing cause-consequence relationships is challenging but would provide deeper mechanistic insights. In particular, conducting rescue experiments to directly relate transcriptional changes with the H3.3K9A mutation would be important.

Response: As the reviewer pointed out, the extent of transcriptional dysregulation in H3.3K9A mESCs exceeds that in H3.3 knockout mESCs. The main difference from the H3.3/chaperones KO approach is that with the presence of H3.3K9A, the incorporation of the histone variant into chromatin remains largely similar to that observed in the wild-type condition. In KO approaches, the complete absence of the H3.3 variant is likely compensated by the deposition of canonical histone H3. We have added this point to the “Discussion” (pages 20-21).

In response to the reviewer's suggestion, we conducted rescue experiments to investigate the direct relationship between transcriptional activation (e.g., immune-related genes, developmental genes, ERVs) and the H3.3K9A mutation by introducing H3 constructs (H3.1, H3.3, H3.3K9A) into the H3.3K9A mESCs (Rebuttal Fig. 1). In brief, we ectopically expressed different H3 constructs in the H3.3K9A ESCs using the PiggyBac system, generated multiple clonal lines as biological replicates, and assessed target gene expressions via RT-qPCR. We observed a specific rescue effect with H3.3, whereby the aberrant activation of immune-associated genes (Fig. 4j), developmental genes (Fig. S11), and ERVs (Fig. 5j) in H3.3K9A mESCs reduced upon ectopic expression of H3.3, but not with H3.1 nor H3.3K9A. Furthermore, when we assessed H3K9me3 levels using ChIP-qPCR, this reduction in aberrant gene activation was accompanied by an increased H3K9me3 at the corresponding putative cis-regulatory regions identified through the GRN, supporting a direct impact of H3.3 on these cis-regulatory regions. The aberrant increase in CD59a protein levels in the H3.3K9A mutant also decreased upon ectopic expression of H3.3 (Fig. 4k). Taken together, these new results (new panels in Fig. 4i,j,k; 5i,j; S11f,g and reported in Rebuttal Fig. 1 below) support a direct role of H3.3K9 in the tested set of transcriptional changes.

Immune-related genes (new Fig. 4i-k)

Developmental genes (new Fig. S11f-g)

ERVs (new Fig. 5c and 5i)

Rebuttal Figure 1: Rescue experiments in K9A mESCs.

In parallel, we analyzed the transcriptional increases in immune-associated genes further. This increase may originate from the activation of cryptic enhancers in ERV-linked cis-regulatory regions and/or the activation of ERV transcripts that subsequently induce an

innate immune response. To test the latter possibility (viral mimicry of ERV transcripts), we treated mESCs with a RIG-I inhibitor (RIG012) [9], which blocks the RIG-I innate immune receptor pathway triggered by ERV transcripts. Treatment with 2.5 μ M RIG012, a concentration that effectively reduces RIG-I activity in cells [9], induced mild toxicity (about 80% viable mESCs) across all mESC lines. RT-qPCR results treating mESC lines with 2.5 μ M RIG012 for 5 hours showed sustained upregulation of immune genes in K9A mutants, comparable to activation observed with the DMSO control, without affecting pluripotency-related genes such as Pou5f1, Sox2, and Klf4 genes (see new Figure S8b and Rebuttal Fig. 2 below). These findings support a predominant role of cryptic enhancer activation and align with previous reports of underdeveloped innate immune responses in mESCs [10, 11].

RT-qPCR upon RIG inhibitor treatment (new Fig. S8b)

Rebuttal Figure 2: RIG-I inhibitor experiment.

Additional points:

* It is not clear if independent mutant clones were analyzed, if they exhibit the same phenotype, and if the experiments were replicated using independent clones.

Response: We apologize if the mention of three independent mutant clones was unclear. To emphasize this point, we modified a sentence at the beginning of the “Results” section (page 3) as follows:

“For each mutant, we generated three independent biological replicates, and after validating the chromosomal integrity of the edited lines (Fig. S1f), we assessed changes in the global levels of histone modifications and gene expression.”

For clarity, we consistently utilized three independent control clones, three independent H3.3K9A clones, and three independent H3.3K27A clones to generate our genomic data (mRNA-seq, ChIP-seq, and PRO-seq), mESC phenotypic assays, and cell differentiation experiments, including cardiomyocyte, neuronal, and macrophage differentiation (new Figure 4e-g). We observed consistent changes across three biological replicates. For the PiggyBac rescue experiment, we utilized all three biological replicates of H3.3K9A mESCs and conducted the rescue independently with the three different constructs in each clone (hence, generating nine new mESC lines).

* The deposited NGS data could not be accessed, making it impossible to assess the data quality.

Response: We apologize for the deposited NGS data not being accessible. We have ensured that the provided links are functional after double-checking with the ArrayExpress support team. Reviewers can now access the datasets before we make them public:

- mRNA-seq E-MTAB-12866: <https://tinyurl.com/2ksnwwdx>
- DNA-seq E-MTAB-12867: <https://tinyurl.com/2p8pwt3f>
- ChIP-seq E-MTAB-12868: <https://tinyurl.com/2p8sv958>
- PRO-seq E-MTAB-12869: <https://tinyurl.com/4dndk9xz>

We also provided sequencing depth, peak calling parameters, and data quality information in the reporting summary file.

* It would be beneficial to include more IGV screenshots to gain a better impression of the data, including H3.3 tracks.

Response: We appreciate the reviewer’s request to include H3.3 tracks in the genome browser screenshots. Thanks to the suggestion, we noticed subtle locus-specific differences in H3.3 levels at certain genomic regions. However, the overall H3.3 enrichment was not significantly altered at Cluster 1/Cluster 2 H3K9me3 regions (Fig. 3c) as well as at ERVs (Fig. 5e). We have updated IGV screenshots including H3.3 tracks and provided three additional examples, which can be found in the new panels of Fig. 6a-b, 7i, and in the new Supplementary Figure 9 and 11 (reported in Rebuttal Fig. 3 below).

Rebuttal Figure 3: Old and additional genome browser shots with H3.3 tracks.

* Figure S4d shows gene dysregulation in the vicinity of H3K9me3 target loci. It appears that there is significant regulation in the non-changed H3K9me3 regions. This finding

could actually support the notion that indirect effects are major contributors to the observed transcriptional changes.

Response: The reviewer is correct. There are transcriptional changes that we interpret as secondary effects. Specifically, the observed gene expression changes in Fig. S4d align with the slight yet significantly increased transcriptional activity at distal non-differentially abundant H3K9me3 regions, measured by PRO-seq (Fig. 3e), and are consistent with the slight increase in H3K27ac (Fig. 3c). These non-DA H3K9me3 regions do not exhibit high H3.3 enrichment (Fig. 3c). Thus, we think that these indirect gene expression changes reflect the profound rewiring of transcriptional programs in K9A mESCs.

Among the gene expression changes detected, the construction of a GRN with the GRANIE package [12] allowed us to retrieve a more refined set of putative enhancer-gene connections. Moreover, our rescue experiments clarified that for the examined loci, changes in the H3K9me3 signal at differentially regulated regions directly influenced the expression of potential target genes.

* H3.3K27A knockout cells were previously published by the Noh lab. Are these the same cells?

Response: No. In previously published work (doi:10.1039/D1MO00352F [13]), we generated a single H3.3K27A clone using the plasmid-encoded Cas9 (pSpCas9(BB)-2A-GFP; PX458 vector), which a collaborator lab used for proteomic studies.

We utilized a CRISPR RNP-based method for the current study to enhance editing efficiency and minimize off-target effects. We generated three new H3.3K27A clones (i.e., the three independent H3.3K27A clones mentioned above) in addition to the three new H3.3K9A clones and used them thoroughly for the experiments presented here.

Reviewer #3 (Remarks to the Author):

To investigate the role of H3.3 K9 and K27 PTM, authors mutated these two residues in mouse mESCs and characterized the epigenomic landscape of these cells.

Taking advantage of an initial RNA-seq experiment, they identified GO categories of differentially expressed genes. Importantly, upregulated genes in K9A were related mainly to immune response while upregulated genes in K27A and in both K9A and K27A mutants were commonly enriched for development and differentiation processes.

Noting that some of the genes were known markers of embryonic lineages, they applied in vitro differentiation protocols to find that K9A mESCs failed to differentiate in neurons, induced cardiac-mesoderm EBs failed to progress further differentiation and cells showed increased apoptosis.

They then carried out ChIP-seq experiments identifying differential peaks that could be clustered in two groups:

Cluster 1 lost the H3K9me3 signal in the K9A mutant, presented a higher basal level of H3K27ac in control mESCs, which was further increased in K9A mESCs. It showed a marked gain in nascent transcription and a relative enrichment of transcriptional co-activators, subunits of chromatin remodelers and TFs linked to pluripotency and development.

Cluster 2 had no basal histone H3 acetylation, a modest increase in H3K27ac and H3K9ac in K9A mESCs and a relatively higher enrichment for core heterochromatin factors, such as HP1 proteins and other transcriptional repressors.

They then showed that a significant fraction of the cryptic CREs de-repressed in H3.3K9A mESCs, function as enhancers active in specialized immune cell types. Importantly, a large fraction of the cryptic CREs activated in H3.3K9A mESCs is ERV-derived enhancers, and some of them are capable of driving the expression of immune-response genes.

Finally, they propose a model where in K9A mESCs, de-repressed ERV-derived CREs upregulate a subset of genes and potentially provide a plethora of TFs binding sites resulting in a redistribution of TFs and subsequent gene expression changes.

The paper is interesting and addresses a very important biological problem in the field. Experiments and data analysis are well done and results are compelling.

Response: We thank the reviewer for the positive note and for acknowledging the relevance of our study.

There are however some crucial points that remain unclear and need to be addressed.

First of all, the paper ignores what is already known about the role of ERV in ESCs. There are several papers that have studied this important issue (see i.e. ; doi:10.1038/nsmb.2799; doi:10.1038/ng.600; doi:10.1038/ng.2965). Please analyze and discuss these data under the light of previous knowledge.

Response: The reviewer raised an important point regarding the overall activation and function of ERVs in wild-type mESCs, which we overlooked in our initial submission. It is important to elaborate on the known function of ERVs in mESCs. In the revised manuscript, we addressed this point in the “Results” (see the next points below) and in the “Discussion” sections. Specifically, we included the following paragraph in the discussion (pages 20-21), and we appreciate the reviewer for bringing it to our attention.

“Previous studies reported the expression of specific ERV subfamilies in mouse and human pluripotent stem cells [14–17], particularly ERVK and MaLR families in mESCs, which harbor pluripotency TF binding sites and act as CREs involved in early embryonic gene expression program [15, 16]. Some ERVs in mESCs have been shown to display both repressive H3K9me3 and active H3K27ac histone marks [6]. In alignment with this, our study identified three distinct groups of ERVs based on their initial activity level and response to the H3.3K9A mutation: those marked by H3K27ac in control mESCs and maintained active in K9A mutants, those exhibiting H3K9me3 and H3K27ac marks in control mESCs and further activated in K9A mutants, and those typically inactive with

H3K9me3 mark in control mESCs but activated in K9A mutants. The activity of these ERVs in mESCs appears to be controlled by a delicate interplay between repressive and activating chromatin signals. Disrupting this balance in H3.3K9A mESCs led to excessive activation of ERVs, highlighting the critical role of H3.3K9 in governing ERVs and varying their baseline activity levels in mESCs. Future studies investigating the function of H3.3K9 mediated ERV dynamics in somatic cells will be a significant area of research.”

What is happening to the expression of ERV elements that are believed to be expressed and functional in ESC? What is the interplay, commonalities and differences between ERV involved in ESC maintenance and differentiation in immune cell types? Any difference in TFBS?

Response: We appreciate the reviewer for raising this point. Previous studies by Fort & Hashimoto *et al.* (2014) [16], Sundaram *et al.* (2017) [15], and others have reported the predominant expression of ERVs from the ERVK and ERVL-MaLR families in mESCs. Our analysis of ERV expression using the RepEnrich2 tool [18] corroborated these findings, demonstrating higher expression levels of ERVK and ERVL-MaLR families on average in control mESCs (a new panel added in Fig. S7c). With the H3.3K9A mutation, we observed a notable increase in expression primarily among ERVK family members, aligning with the enrichment pattern of the H3.3 histone variant, where 70% of significantly enriched LTR elements are ERVK (Fig. 5d). We did not observe a notable activation of ERVL-MaLR elements in K9A mESCs, consistent with the minor fraction (<4%) of this family showing significant H3.3 enrichment.

To assess disparities in TF binding sites between ERVs implicated in mESCs maintenance and those linked to immune gene regulation, we conducted Homer motif enrichment analysis using distinct ERV subsets. As outlined previously, we discerned (i) a group of ERVs already active in the basal state of control mESCs, further activated in K9A mutant, and (ii) another group of ERVs exclusively activated in K9A mutant mESCs. For TF motif enrichment analysis, we retrieved from RepeatMasker the genomic coordinates of each repeat instance for the selected ERV subfamilies in the first/second group. We utilized the resulting BED files for Homer de-novo motif analysis. Analysis of the first group of ERVs (active in control and further activated in K9A) revealed enrichment primarily for transcription factor motifs associated with pluripotency maintenance, such as POU5F1 and POU family members, SOX2, ETS family members, and TCF factors. Apart from pluripotency-related motifs, we observed the enrichment of TF motifs related to inflammation (e.g., NFkB, MITF, and MAFK motifs). Conversely, analysis of ERVs activated only in K9A mESCs exhibited enrichment for TF motifs linked to development and differentiation, such as HOX, FOX, TBX, and ZBTB family members, as well as motifs associated with immune system processes like MAFK, REL/RELA and SPIC/ELF motifs. Of note, some inflammation-related TF motifs were also enriched among ERVs already active in control mESCs.

The results of these analyses are now included in Supplementary Figure S7k-l (and reported in Rebuttal Fig. 4 below).

Rebuttal Figure 4: TF motif analysis for different ERV subsets.

In general what is missing is a general characterization of Transposable elements expression in any of the data analysis that has been carried out in the paper. What is the basic level of expression of TE in general and ERV in particular in WT cells and what is their expression in edited cells? Please identify TE families that are differentially regulated by using ad hoc tools i.e. Square.

Response: We thank the reviewer for this comment. We have extensively revised the result section “Cryptic enhancers in H3.3K9A mESCs derive from ERVs” in response. Specifically, we expanded the original Fig. 5 into two new figures (main Fig. 5 and Supplementary Fig. 7) dedicated to TE analysis. All analyses aimed at elucidating TE activity in both controls and mutants were conducted using ad-hoc tools. For TE expression analysis, encompassing non-LTRs and LTRs, we utilized the latest release of RepEnrich (Criscione *et al.*, 2014 [18] - <https://github.com/nerettilab/RepEnrich2>). For ChIP-seq data analysis at TE sequences, we employed T3E (Transposable Element Enrichment Estimator [19]), a recently developed computational tool that allows for precise categorization of ERV subfamilies based on histone modification patterns.

To profile TE expression of non-LTR in control, K27A, and K9A mESCs, we retrieved non-LTR annotation from RepeatMasker and utilized the RepEnrich2 tool. Overall, LINE and SINE elements showed higher expression in control mESCs compared to LTRs (mean log-transformed RNA normalized counts LINE=2.76, SINE=4.03, LTR=2.26). While we observed a trend towards upregulation for LINE and downregulation for SINE elements from the global comparison of count values with K27A and K9A mESCs, these changes were not significant (P-value LINE_K9AvsCtrl=0.8, SINE_K9AvsCtrl=0.65; new panel in Supplementary Fig. 7a). Differential expression analysis of non-LTRs conducted with DESeq2 revealed mild changes in the expression of few specific LINEs and SINEs in K9A mESCs, but no particular changes were observed in K27A mESCs (new panel in Supplementary Fig. 7b).

We also expanded the transcriptomic analysis of LTR elements using the RepEnrich2 tool to profile the basal expression levels of ERV families in control mESCs, confirming the globally high expression levels of ERVK and MaLR families (discussed in the previous

point and shown in the new panel in Fig. S7c). The PCA and volcano plots with differential expression analysis of ERVs indicated a limited impact of the H3.3K27A mutation (Fig. S7d,e). However, the differential expression analysis revealed extensively increased transcriptional activity of specific ERV subfamilies in K9A mESCs, including members of the ERVK family, such as IAP, IAPE, MMERVK, and RLTR elements (Fig. 5b).

Regarding the ChIP-seq analysis at repeats conducted with T3E, we performed a detailed examination of histone modifications at ERVs. This enabled us to stratify ERV subfamilies based on their basal activity levels and changes in histone marks upon the introduction of the H3.3K9A mutation in mESCs. Specifically, as described earlier, we identified three groups of ERVs with distinct patterns of histone modifications. The first group showed non-significant H3K9me3 and significant H3K27ac levels in control mESCs, which remained unchanged in K9A mESCs (new panel in Supplementary Fig. 7j). The second group exhibited H3K9me3 and H3K27ac marking in control mESCs, with a decrease in H3K9me3 and an increase in H3K27ac observed in K9A mESCs. The third group comprised ERVs marked only by H3K9me3 in control mESCs, showing a reduction in H3K9me3 and a gain in H3K27ac in K9A mESCs. Consistent with the transcriptomic analysis, ERVs belonging to the third group represented the majority of the subfamilies identified by the differential expression analysis (see first cluster of the heatmap in Fig. 5h).

To highlight the alterations in chromatin marks at ERVs in control and edited mESCs, we moved the heatmap summarizing the H3.3, H3K9me3, and H3K27ac ChIP-seq analyses at ERVs from Supplementary to the main Fig. 5 (new panel in Fig. 5h). This heatmap provides a comprehensive overview of the chromatin changes at the ERV subfamilies exhibiting the highest H3.3 enrichment over input.

With the newly generated K9A mESC lines overexpressing histone H3 constructs for the rescue experiments, we performed H3K9me3 ChIP-qPCR and RT-qPCR to monitor the expression of ERV sequences. These new experiments revealed a significant increase in the repressive H3K9me3 mark at IAP sequences only in K9A mESCs expressing WT H3.3. This increase in H3K9me3 was correlated with decreased expression, thereby confirming the central role of the H3.3 lysine 9 residue in ERV silencing.

We have updated Fig. 5 and Supplementary Fig. 7 to incorporate these new findings and have revised the main text accordingly (pages 12-13).

Rebuttal Figure 5a: Updated version of main Figure 5.

Rebuttal Figure 5b: Updated version of Supplementary Figure 7.

How much of your H3K9me3 ChIP-seq signal is unmappable?

Response: Approximately 50% of the H3K9me3 ChIP-seq reads did not map uniquely. In our ChIP-seq analyses at TEs, we included these multi-mapping reads as the T3E computational tool internally manages ambiguously mapped reads, ensuring a comprehensive examination (see Almeida da Paz & Laher, 2022 [19] for details).

Given the experience in differentiating cell lineages in authors' laboratory and the knowledge accumulated in this study, an experiment of *in vitro* differentiation of functional immune cell types should be required as a final validation of the functional implications of reactivation of cryptic enhancers for the immune system.

Response: We appreciate this suggestion. To address it, we conducted an *in-vitro* differentiation of mESCs into macrophages using a protocol published by Zhuang *et al.*, 2012 [20] (see updated Methods section for details). Briefly, mESCs were prompted to differentiate into macrophages by exposure to a differentiation medium containing interleukin 3 (IL-3) and conditioned medium from L929 cells providing macrophage colony-stimulating factor (M-CSF/CSF-1). At the initial stages of the differentiation, RT-qPCR analysis revealed an increased expression of early hematopoietic and macrophage markers in K9A embryoid bodies (day 8; new panel in Fig. 4f). However, the K9A EBs exhibited irregular morphology compared to control EBs (new Supplementary Fig. 6a). This premature expression of marker genes did not persist in later stages of differentiation (day 12; new panel in Fig. 4g), although K9A EBs were able to form Factories similar to controls when transferred to gelatin-coated dishes (Fig. S6a). A striking difference was observed at the macrophage-precursors stage (Factory-derived suspension cells after day 12), where over 90% of cells derived from K9A Factories died and failed to produce terminally differentiated macrophages upon further culture in macrophage medium (Fig. 4h and Fig. S6a). In contrast, control cells were able to generate terminally differentiated macrophages. Immunolabeling of the control ESCs-derived macrophages indicated a high percentage of Cd11b/F4-80 double-positive cells, thus confirming the effectiveness of the differentiation protocol.

This result supports a model that activating *cis*-regulatory elements associated with immune genes could initiate the differentiation of K9A mESCs toward immune-cell lineages, leading to a premature expression of early hematopoietic and macrophage marker genes. However, similar to observations in neuronal and cardiomyocyte differentiations, the K9A cells fail to generate fully differentiated cells, likely due to the accumulation of regulatory defects in the genome.

We have incorporated these results in Fig. 4 and Supplementary Figure 6. For an overview, a summary panel is shown below in Rebuttal Fig. 6.

In-vitro macrophage differentiation (new Fig. 4f-h and S6)

Staining with anti-CD11b and anti-F4-80 of Ctrl ESDMs (day 20) - FACS gating strategy

Rebuttal Figure 6: In-vitro macrophage differentiation.

A more detailed description of TFs linked to pluripotency in Cluster 1 should be presented.

Response: We thank the reviewer for raising this point. To provide a more detailed description of the TF motifs enriched at Cluster 1 H3K9me3 regions, we selected active regions identified through the dREG package that fell within Cluster 1 and performed de novo motif enrichment analysis using Homer (with “-size 200” parameter, to take the center of PRO-seq peak). Among the top-enriched TF motifs, we observed several motifs associated with pluripotency maintenance, including POU5F1, SOX2/SOX9, RARA, and the OCT4::SOX2::TCF::NANOG (OSTN) consensus motif (new panel in Fig. 6c). This result aligns with the CHIP-Atlas data integration presented in Fig. 3d, which

demonstrated a relative enrichment of pluripotency-related TFs at Cluster 1 regions. As expected, overlapping TF motifs were also identified at “strong” canonical enhancers (new panel in Fig. 6d).

The new analyses of TF motif enrichment are included in the revised version of Fig. 6, and we revised the main text accordingly (page 15).

For comparison, we also used the same approach and performed a TF motifs analysis for Cluster 2 regions. We identified motifs such as FOXN3, YY1, ZBTB12, IRF4/6, MX11, NKX, CDX2, and SPIC, among others. This result is consistent with the TF motifs enriched at ERVs activated only in K9A mESCs (presented above and in Fig. S7I) because about 81% (n=853/1054) of the repeat instances of subfamilies belonging to this group were located within Cluster 2 regions. With regard to non-DA H3K9me3 regions, we mostly observed motifs associated with transcriptional repressors such as OSR1/2, ZEB1/2, and several ZFPs.

TF motif analysis at Cluster 1 H3K9me3 regions (new Fig. 6c-d)

Rebuttal Figure 7: TF motifs analysis at Cluster 1 (and Cluster 2) H3K9me3 regions, and at canonical enhancers.

References

1. Elsässer SJ, Noh K-M, Diaz N, Allis CD, Banaszynski LA (2015) Histone H3.3 is required for endogenous retroviral element silencing in embryonic stem cells. *Nature* 522:240–244
2. Sadic D, Schmidt K, Groh S, Kondofersky I, Ellwart J, Fuchs C, Theis FJ, Schotta G (2015) Atrx promotes heterochromatin formation at retrotransposons. *EMBO Rep* 16:836–850
3. Wolf G, Rebollo R, Karimi MM, et al (2017) On the role of H3.3 in retroviral silencing. *Nature* 548:E1–E3
4. Groh S, Schotta G (2017) Silencing of endogenous retroviruses by heterochromatin. *Cell Mol Life Sci* 74:2055–2065
5. Rowe HM, Jakobsson J, Mesnard D, et al (2010) KAP1 controls endogenous retroviruses in embryonic stem cells. *Nature* 463:237–240
6. He J, Fu X, Zhang M, et al (2019) Transposable elements are regulated by context-specific patterns of chromatin marks in mouse embryonic stem cells. *Nat Commun* 10:34
7. Rowe HM, Friedli M, Offner S, Verp S, Mesnard D, Marquis J, Aktas T, Trono D (2013) De novo DNA methylation of endogenous retroviruses is shaped by KRAB-ZFPs/KAP1 and ESET. *Development* 140:519–529
8. Rowe HM, Kapopoulou A, Corsinotti A, Fasching L, Macfarlan TS, Tarabay Y, Viville S, Jakobsson J, Pfaff SL, Trono D (2013) TRIM28 repression of retrotransposon-based enhancers is necessary to preserve transcriptional dynamics in embryonic stem cells. *Genome Res* 23:452–461
9. Rawling DC, Jagdmann GE Jr, Potapova O, Pyle AM (2020) Small-Molecule Antagonists of the RIG-I Innate Immune Receptor. *ACS Chem Biol* 15:311–317
10. Wang R, Wang J, Paul AM, Acharya D, Bai F, Huang F, Guo Y-L (2013) Mouse Embryonic Stem Cells Are Deficient in Type I Interferon Expression in Response to Viral Infections and Double-stranded RNA*. *J Biol Chem* 288:15926–15936
11. Wang R, Wang J, Acharya D, Paul AM, Bai F, Huang F, Guo Y-L (2014) Antiviral Responses in Mouse Embryonic Stem Cells: Implications for Targeting G4 DNA as a novel therapeutic approach*. *J Biol Chem* 289:25186–25198
12. Kamal A, Arnold C, Claringbould A, et al (2023) GRaNIE and GRaNPA: inference and evaluation of enhancer-mediated gene regulatory networks. *Mol Syst Biol* 19:e11627
13. Kori Y, Lund PJ, Trovato M, Sidoli S, Yuan Z-F, Noh K-M, Garcia BA (2022) Multi-omic profiling of histone variant H3.3 lysine 27 methylation reveals a distinct role from canonical H3 in stem cell differentiation. *Mol Omics* 18:296–314
14. Kunarso G, Chia NY, Jeyakani J, Hwang C, Lu X, Chan YS, Ng HH, Bourque G (2010) Transposable elements have rewired the core regulatory network of human embryonic stem cells. *Nat Genet* 42:631–634
15. Sundaram V, Choudhary MNK, Pehrsson E, et al (2017) Functional cis-regulatory modules encoded by mouse-specific endogenous retrovirus. *Nat Commun* 8:14550
16. Fort A, Hashimoto K, Yamada D, et al (2014) Deep transcriptome profiling of mammalian stem cells supports a regulatory role for retrotransposons in pluripotency maintenance. *Nat Genet* 46:558–566
17. Lu X, Sachs F, Ramsay L, Jacques P-É, Göke J, Bourque G, Ng H-H (2014) The retrovirus HERVH is a long noncoding RNA required for human embryonic stem cell identity. *Nat Struct Mol Biol* 21:423–425
18. Criscione SW, Zhang Y, Thompson W, Sedivy JM, Neretti N (2014) Transcriptional landscape of repetitive elements in normal and cancer human cells. *BMC Genomics* 15:1–17
19. Almeida da Paz M, Taher L (2022) T3E: a tool for characterising the epigenetic profile of transposable elements using ChIP-seq data. *Mob DNA* 13:29
20. Zhuang L, Pound JD, Willems JJLP, Taylor AH, Forrester LM, Gregory CD (2012) Pure populations of murine macrophages from cultured embryonic stem cells. Application to studies of chemotaxis and apoptotic cell clearance. *J Immunol Methods* 385:1–14

REVIEWERS' COMMENTS

Reviewer #1 (Remarks to the Author):

The authors have improved the manuscript by providing new data and analyses. In particular, they performed rescue experiments, demonstrating that wild type H3.3 can rescue impaired silencing of selected ERVs and developmental genes. This suggests specificity of the phenotype on such elements and increases confidence in the data. They also performed RIG inhibitor treatment and excluded the possibility that viral mimicry is causing up-regulation of immune-related genes. In addition, they provide TF binding predictions that suggest distinct TFs to be responsible for observed transcriptional changes.

All this improves the manuscript, but did not bring it to a new mechanistic level. It is still largely descriptive, but certainly provides interesting information on the role of H3.3K9A that is relevant to understand heterochromatin dynamics and ERV regulation.

Due to the complexity of the phenotype (especially H3.3K9A), I would definitely add to the discussion and/or provide a "limits of the study" section:

It needs to be made clear that H3.3 is not exclusively associated with heterochromatin. It is also important in the context of active transcription. H3.3 can be deposited by the Daxx/Atrx/Morc3 pathway in heterochromatin (associated with H3K9me3) and by Hira in the context of euchromatin/active transcription. In the context of Daxx, H3.3 appears to be pre-modified with K9me3 before deposition. Failure to do this in the K9A mutant seems to be compatible with deposition, but it is unclear if mediated by Daxx or Hira. Transcriptionally active ERVs may actually have switched to Hira deposition as consequence of their transcriptional activity and the switch in chromatin state. This would be compatible with changes in the H3.3 pattern on such loci.

Due to the deposition of H3.3 in euchromatic regions, indirect effects of H3.3K9A may result from impaired H3K9ac (or me1/2) on such regions. The consequences are unclear, but a likely expectation would be reduced enhancer/promoter activity on sensitive loci.

minor point:

It is actually quite annoying that data access was not resolved. I could open the indicated URLs and could see that data sets were deposited, but download access was not enabled, so I could not check data quality. In general, I would urge for providing the relevant bigwig files and count matrices for reviewer access (definitely possible with GEO).

Reviewer #2 (Remarks to the Author):

I have deeply appreciated the efforts made by the authors to address my points. I feel that they answered satisfactorily to all my comments.

A minor point:

I would like they include in the main text of the manuscript the important piece of information that "Approximately 50% of the H3K9me3 ChIP-seq reads did not map uniquely".

We thank the reviewers for acknowledging our efforts to improve the manuscript. Based on their suggestions, we provide our point-by-point responses to the new comments.

REVIEWERS' COMMENTS

Reviewer #1 (Remarks to the Author):

The authors have improved the manuscript by providing new data and analyses. In particular, they performed rescue experiments, demonstrating that wild type H3.3 can rescue impaired silencing of selected ERVs and developmental genes. This suggests specificity of the phenotype on such elements and increases confidence in the data. They also performed RIG inhibitor treatment and excluded the possibility that viral mimicry is causing up-regulation of immune-related genes. In addition, they provide TF binding predictions that suggest distinct TFs to be responsible for observed transcriptional changes.

All this improves the manuscript, but did not bring it to a new mechanistic level. It is still largely descriptive, but certainly provides interesting information on the role of H3.3K9A that is relevant to understand heterochromatin dynamics and ERV regulation.

Due to the complexity of the phenotype (especially H3.3K9A), I would definitely add to the discussion and/or provide a "limits of the study" section:

It needs to be made clear that H3.3 is not exclusively associated with heterochromatin. It is also important in the context of active transcription. H3.3 can be deposited by the Daxx/Atrx/Morc3 pathway in heterochromatin (associated with H3K9me3) and by Hira in the context of euchromatin/active transcription. In the context of Daxx, H3.3 appears to be pre-modified with K9me3 before deposition. Failure to do this in the K9A mutant seems to be compatible with deposition, but it is unclear if mediated by Daxx or Hira. Transcriptionally active ERVs may actually have switched to Hira deposition as consequence of their transcriptional activity and the switch in chromatin state. This would be compatible with changes in the H3.3 pattern on such loci.

Due to the deposition of H3.3 in euchromatic regions, indirect effects of H3.3K9A may result from impaired H3K9ac (or me1/2) on such regions. The consequences are unclear, but a likely expectation would be reduced enhancer/promoter activity on sensitive loci.

We thank the reviewer for acknowledging the improvement of our manuscript and providing additional feedback. We agree with stating the limitations of our study as suggested. In response, we have added the following paragraph to the discussion:

Nevertheless, our study does not define the chaperone system responsible for the H3.3K9A mutant deposition. H3.3 is associated with both heterochromatin and active transcription, requiring histone turnover. The DAXX/ATRX/MORC3 chaperone pathway deposits H3.3 in heterochromatin, while HIRA deposits H3.3 in euchromatin/sites of active transcription (PMID: 38297159). In the DAXX-related pathway, H3.3 can be pre-modified with K9me3 before deposition (PMID: 36868228). The H3.3K9A mutant appears compatible with heterochromatin deposition, but whether DAXX or HIRA mediates this process is unclear. Transcriptionally active ERVs may have switched to HIRA deposition due to their transcriptional activity and changes in chromatin state.

minor point:

It is actually quite annoying that data access was not resolved. I could open the indicated URLs and could see that data sets were deposited, but download access was not enabled, so I could not check data quality. In general, I would urge for providing the relevant bigwig files and count matrices for reviewer access (definitely possible with GEO).

We sincerely apologize that the reviewer could not verify the data quality of our sequencing results. To improve data access and facilitate easier verification of our data quality, we have deposited the complete set of BigWig files in the ArrayExpress database.

Furthermore, we have taken steps to make it easier for reviewers to visualize the ChIP-seq and mRNA-seq data used to generate the genome browser snapshots. We have prepared a UCSC Genome Browser Custom Session, which is now available at this link: <https://tinyurl.com/Trovatoetal2024UCSC> .

Reviewer #2 (Remarks to the Author):

I have deeply appreciated the efforts made by the authors to address my points. I feel that they answered satisfactorily to all my comments.

A minor point:

I would like they include in the main text of the manuscript the important piece of information that "Approximately 50% of the H3K9me3 ChIP-seq reads did not map uniquely".

We thank the reviewer for appreciating our efforts. As requested by the reviewer, we added to the "ChIP-seq analysis" paragraph in the Methods section the following sentence:

For H3K9me3 ChIP-seq, approximately 50% of the reads did not map uniquely.

Summary of the main findings:

In this study, the authors mutated histone H3.3 K9 and K27 residues, demonstrating their importance in maintaining repressive chromatin states at endogenous retroviruses-derived cryptic enhancers and bivalent promoters in mouse embryonic stem cells.